# TD-MPC2: Scalable, Robust World Models for Continuous Control

**Nicklas Hansen**[⋆], **Hao Su**[⋆†], **Xiaolong Wang**[⋆†]
[⋆]University of California San Diego, [†]Equal advising
{nihansen,haosu,xiw012}@ucsd.edu

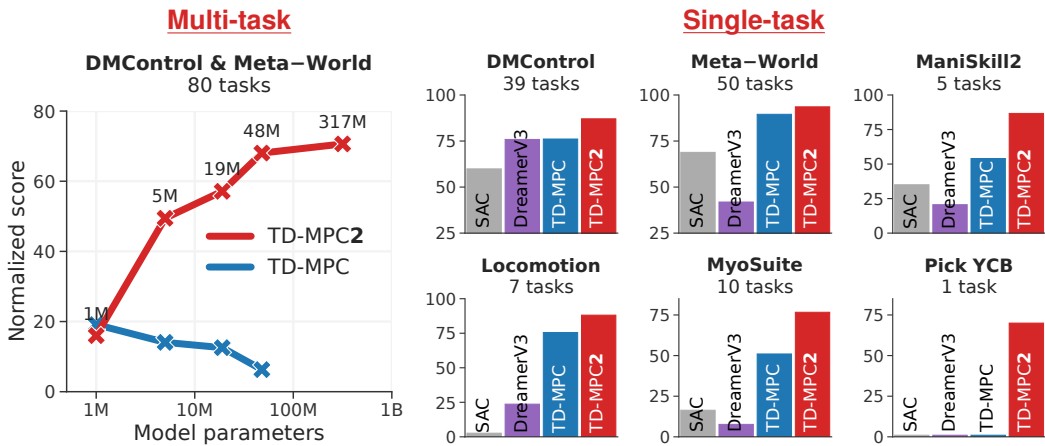

*Figure 1.* **Overview.** TD-MPC2 compares favorably to existing model-free and model-based RL methods across **104** continuous control tasks spanning multiple domains, with a *single* set of hyperparameters (*right*). We further demonstrate the scalability of TD-MPC2 by training a single 317M parameter agent to perform **80** tasks across multiple domains, embodiments, and action spaces (*left*).

## ABSTRACT

TD-MPC is a model-based reinforcement learning (RL) algorithm that performs local trajectory optimization in the latent space of a learned implicit (decoder-free) world model. In this work, we present TD-MPC2: a series of improvements upon the TD-MPC algorithm. We demonstrate that TD-MPC2 improves significantly over baselines across **104** online RL tasks spanning 4 diverse task domains, achieving consistently strong results with a single set of hyperparameters. We further show that agent capabilities increase with model and data size, and successfully train a single 317M parameter agent to perform **80** tasks across multiple task domains, embodiments, and action spaces. We conclude with an account of lessons, opportunities, and risks associated with large TD-MPC2 agents.

**Explore videos, models, data, code, and more at** **https://tdmpc2.com**

## 1 INTRODUCTION

Training large models on internet-scale datasets has led to generalist models that perform a wide variety of language and vision tasks (Brown et al., 2020; He et al., 2022; Kirillov et al., 2023). The success of these models can largely be attributed to the availability of enormous datasets, and carefully designed architectures that reliably scale with model and data size. While researchers have recently extended this paradigm to robotics (Reed et al., 2022; Brohan et al., 2023), a generalist embodied agent that learns to perform diverse control tasks via low-level actions, across multiple embodiments, from large uncurated (*i.e.*, mixed-quality) datasets remains an elusive goal. We argue that current approaches to generalist embodied agents suffer from *(a)* the assumption of near-expert trajectories for behavior cloning which severely limits the amount of available data (Reed et al., 2022; Lee et al., 2022; Kumar et al., 2022; Schubert et al., 2023; Driess et al., 2023; Brohan et al., 2023), and *(b)* a lack of scalable continuous control algorithms that are able to consume large uncurated datasets.

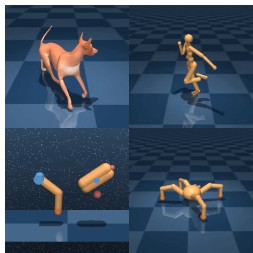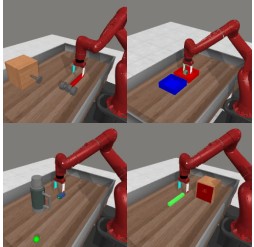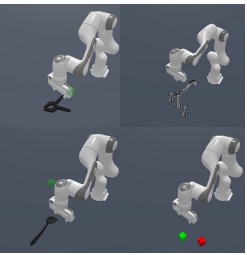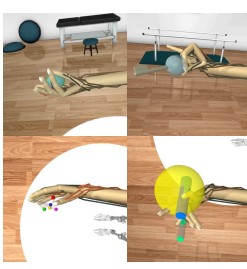

*Figure 2.* **Tasks.** TD-MPC**2** performs **104** diverse tasks from (left to right) DMControl (Tassa et al., 2018), Meta-World (Yu et al., 2019), ManiSkill2 (Gu et al., 2023), and MyoSuite (Caggiano et al., 2022), with a *single* set of hyperparameters. See Appendix B for visualization of all tasks.

Reinforcement Learning (RL) is an ideal framework for extracting expert behavior from uncurated datasets. However, most existing RL algorithms (Lillicrap et al., 2016; Haarnoja et al., 2018) are designed for single-task learning and rely on per-task hyperparameters, with no principled method for selecting those hyperparameters (Zhang et al., 2021). An algorithm that can consume large multi-task datasets will invariably need to be robust to variation between different tasks (*e.g.*, action space dimensionality, difficulty of exploration, and reward distribution). In this work, we present TD-MPC**2**: a significant step towards achieving this goal. TD-MPC**2** is a model-based RL algorithm designed for learning generalist world models on large uncurated datasets composed of multiple task domains, embodiments, and action spaces, with data sourced from behavior policies that cover a wide range of skill levels, and without the need for hyperparameter-tuning.

Our algorithm, which builds upon TD-MPC (Hansen et al., 2022), performs local trajectory optimization in the latent space of a learned implicit (decoder-free) world model. While the TD-MPC family of algorithms has demonstrated strong empirical performance in prior work (Hansen et al., 2022; 2023; Yuan et al., 2022; Yang et al., 2023; Feng et al., 2023; Chitnis et al., 2023; Zhu et al., 2023; Lancaster et al., 2023), most successes have been limited to single-task learning with little emphasis on scaling. As shown in Figure 1, naïvely increasing model and data size of TD-MPC often leads to a net *decrease* in agent performance, as is commonly observed in RL literature (Kumar et al., 2023). In contrast, scaling TD-MPC**2** leads to consistently improved capabilities. Our algorithmic contributions, which have been key to achieving this milestone, are two-fold: *(1)* improved algorithmic robustness by revisiting core design choices, and *(2)* careful design of an architecture that can accommodate datasets with multiple embodiments and action spaces without relying on domain knowledge. The resulting algorithm, TD-MPC**2**, is scalable, robust, and can be applied to a variety of single-task and multi-task continuous control problems using a *single* set of hyperparameters.

We evaluate TD-MPC**2** across a total of **104** diverse continuous control tasks spanning 4 task domains: DMControl (Tassa et al., 2018), Meta-World (Yu et al., 2019), ManiSkill2 (Gu et al., 2023), and MyoSuite (Caggiano et al., 2022). We summarize our results in Figure 1, and visualize task domains in Figure 2. Tasks include high-dimensional state and action spaces (up to $\mathcal{A} \in \mathbb{R}^{39}$), sparse rewards, multi-object manipulation, physiologically accurate musculoskeletal motor control, complex locomotion (*e.g.* Dog and Humanoid embodiments), and cover a wide range of task difficulties. Our results demonstrate that TD-MPC**2** consistently outperforms existing model-based and model-free methods, using the *same* hyperparameters across all tasks (Figure 1, *right*). Here, "Locomotion" and "Pick YCB" are particularly challenging subsets of DMControl and ManiSkill2, respectively. We further show that agent capabilities increase with model and data size, and successfully train a single 317M parameter world model to perform **80** tasks across multiple task domains, embodiments, and action spaces (Figure 1, *left*). In support of open-source science, **we publicly release 300+ model checkpoints, datasets, and code for training and evaluating TD-MPC2 agents, which is available at `https://tdmpc2.com`**. We conclude the paper with an account of lessons, opportunities, and risks associated with large TD-MPC**2** agents.

## 2 BACKGROUND

**Reinforcement Learning** (RL) aims to learn a policy from interaction with an environment, formulated as a Markov Decision Process (MDP) (Bellman, 1957). We focus on infinite-horizon MDPs with continuous action spaces, which can be formalized as a tuple $(\mathcal{S}, \mathcal{A}, \mathcal{T}, R, \gamma)$ where $\mathbf{s} \in \mathcal{S}$ are states, $\mathbf{a} \in \mathcal{A}$ are actions, $\mathcal{T} \colon \mathcal{S} \times \mathcal{A} \mapsto \mathcal{S}$ is the transition function, $\mathcal{R} \colon \mathcal{S} \times \mathcal{A} \mapsto \mathbb{R}$ is a reward function associated with a particular task, and $\gamma$ is a discount factor. The goal is to

derive a control policy $\pi \colon \mathcal{S} \mapsto \mathcal{A}$ such that the expected discounted sum of rewards (return) $\mathbb{E}_\pi \left[ \sum_{t=0}^\infty \gamma^t r_t \right]$, $r_t = R(\mathbf{s}_t, \pi(\mathbf{s}_t))$ is maximized. In this work, we obtain $\pi$ by learning a *world model* (model of the environment) and then select actions by planning with the learned model.

**Model Predictive Control** (MPC) is a general framework for model-based control that optimizes action sequences $\mathbf{a}_{t:t+H}$ of finite length such that return is maximized (or cost is minimized) over the time horizon $H$, which corresponds to solving the following optimization problem:

$$\pi(\mathbf{s}_t) = \arg \max_{\mathbf{a}_{t:t+H}} \mathbb{E} \left[ \sum_{i=0}^H \gamma^{t+i} R(\mathbf{s}_{t+i}, \mathbf{a}_{t+i}) \right] . \tag{1}$$

The return of a candidate trajectory is estimated by simulating it with the learned model (Negenborn et al., 2005). Thus, a policy obtained by Equation 1 will invariably be a (temporally) *locally* optimal policy and is not guaranteed (nor likely) to be a solution to the general reinforcement learning problem outlined above. As we discuss in the following, TD-MPC**2** addresses this shortcoming of local trajectory optimization by bootstrapping return estimates beyond horizon $H$ with a learned terminal value function.

## 3 TD-MPC**2**

Our work builds upon TD-MPC (Hansen et al., 2022), a model-based RL algorithm that performs local trajectory optimization (planning) in the latent space of a learned implicit world model. TD-MPC**2** is a practical algorithm for training massively multitask world models. Specifically, we propose a series of improvements to the TD-MPC algorithm, which have been key to achieving strong algorithmic robustness (can use the same hyperparameters across all tasks) and scaling its world model to $300\times$ more parameters than previously. We introduce the TD-MPC**2** algorithm in the following, and provide a full list of algorithmic improvements in Appendix A.

### 3.1 LEARNING AN IMPLICIT WORLD MODEL

Learning a generative model of the environment using a reconstruction (decoder) objective is tempting due to its rich learning signal. However, accurately predicting raw future observations (*e.g.*, images or proprioceptive features) over long time horizons is a difficult problem, and does not necessarily lead to effective control (Lambert et al., 2020). Rather than explicitly modeling dynamics using reconstruction, TD-MPC**2** aims to learn a *maximally useful* model: a model that accurately predicts *outcomes* (returns) conditioned on a sequence of actions. Specifically, TD-MPC**2** learns an *implicit*, control-centric world model from environment interaction using a combination of joint-embedding prediction (Grill et al., 2020), reward prediction, and TD-learning (Sutton, 1998), *without* decoding observations. We argue that this alternative formulation of model-based RL is key to modeling large datasets with modest model sizes. The world model can subsequently be used for decision-making by performing local trajectory optimization (planning) following the MPC framework.

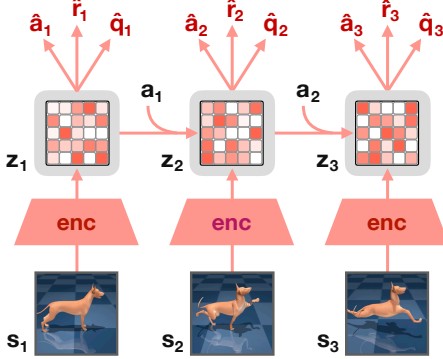

*Figure 3.* **The TD-MPC2 architecture.** Observations $\mathbf{s}$ are encoded into their (normalized) latent representation $\mathbf{z}$. The model then recurrently predicts actions $\hat{\mathbf{a}}$, rewards $\hat{r}$, and terminal values $\hat{q}$, *without* decoding future observations.

**Components.** The TD-MPC**2** architecture is shown in Figure 3 and consists of five components:

| | | | |
|---|---|---|---|
| Encoder | $\mathbf{z} = h(\mathbf{s}, \mathbf{e})$ | ▷ Maps observations to their latent representations | |
| Latent dynamics | $\mathbf{z}' = d(\mathbf{z}, \mathbf{a}, \mathbf{e})$ | ▷ Models (latent) forward dynamics | |
| Reward | $\hat{r} = R(\mathbf{z}, \mathbf{a}, \mathbf{e})$ | ▷ Predicts reward $r$ of a transition | (2) |
| Terminal value | $\hat{q} = Q(\mathbf{z}, \mathbf{a}, \mathbf{e})$ | ▷ Predicts discounted sum of rewards (return) | |
| Policy prior | $\hat{\mathbf{a}} = p(\mathbf{z}, \mathbf{e})$ | ▷ Predicts action $\mathbf{a}^*$ that maximizes $Q$ | |

where $\mathbf{s}$ and $\mathbf{a}$ are states and actions, $\mathbf{z}$ is the latent representation, and $\mathbf{e}$ is a learnable task embedding for use in multitask world models. For visual clarity, we will omit $\mathbf{e}$ in the following unless it is

particularly relevant. The policy prior $p$ serves to guide the sample-based trajectory optimizer (planner), and to reduce the computational cost of TD-learning. During online interaction, TD-MPC**2** maintains a replay buffer $\mathcal{B}$ with trajectories, and iteratively *(i)* updates the world model using data sampled from $\mathcal{B}$, and *(ii)* collects new environment data by planning with the learned model.

**Model objective.** The $h, d, R, Q$ components are jointly optimized to minimize the objective

$$\mathcal{L}(\theta) \doteq \mathbb{E}_{(\mathbf{s}, \mathbf{a}, r, \mathbf{s}')_{0:H} \sim \mathcal{B}} \left[ \sum_{t=0}^{H} \lambda^t \left( \underbrace{\| \mathbf{z}'_t - \mathrm{sg}(h(\mathbf{s}'_t)) \|_2^2}_{\text{Joint-embedding prediction}} + \underbrace{\mathrm{CE}(\hat{r}_t, r_t)}_{\text{Reward prediction}} + \underbrace{\mathrm{CE}(\hat{q}_t, q_t)}_{\text{Value prediction}} \right) \right], \quad (3)$$

where sg is the `stop-grad` operator, $(\mathbf{z}'_t, \hat{r}_t, \hat{q}_t)$ are defined in Equation 2, $q_t \doteq r_t + \gamma \bar{Q}(\mathbf{z}'_t, p(\mathbf{z}'_t))$ is the TD-target at step $t$, $\lambda \in (0, 1]$ is a constant coefficient that weighs temporally farther time steps less, and CE is the cross-entropy. $\bar{Q}$ used to compute the TD-target is an exponential moving average (EMA) of $Q$ (Lillicrap et al., 2016). As the magnitude of rewards may differ drastically between tasks, TD-MPC**2** formulates reward and value prediction as a discrete regression (multi-class classification) problem in a log-transformed space, which is optimized by minimizing cross-entropy with $r_t, q_t$ as soft targets (Bellemare et al., 2017; Kumar et al., 2023; Hafner et al., 2023).

**Policy objective.** The policy prior $p$ is a stochastic maximum entropy (Ziebart et al., 2008; Haarnoja et al., 2018) policy that learns to maximize the objective

$$\mathcal{L}_p(\theta) \doteq \mathbb{E}_{(\mathbf{s}, \mathbf{a})_{0:H} \sim \mathcal{B}} \left[ \sum_{t=0}^{H} \lambda^t \left[ \alpha Q(\mathbf{z}_t, p(\mathbf{z}_t)) - \beta \mathcal{H}(p(\cdot|\mathbf{z}_t)) \right] \right], \ \mathbf{z}_{t+1} = d(\mathbf{z}_t, \mathbf{a}_t), \ \mathbf{z}_0 = h(\mathbf{s}_0), \ (4)$$

where $\mathcal{H}$ is the entropy of $p$ which can be computed in closed form. Gradients of $\mathcal{L}_p(\theta)$ are taken wrt. $p$ only. As magnitude of the value estimate $Q(\mathbf{z}_t, p(\mathbf{z}_t))$ and entropy $\mathcal{H}$ can vary greatly between datasets and different stages of training, it is necessary to balance the two losses to prevent premature entropy collapse (Yarats et al., 2021). A common choice for automatically tuning $\alpha, \beta$ is to keep one of them constant, and adjusting the other based on an entropy target (Haarnoja et al., 2018) or moving statistics (Hafner et al., 2023). In practice, we opt for tuning $\alpha$ via moving statistics, but empirically did not observe any significant difference in results between these two options.

**Architecture.** All components of TD-MPC**2** are implemented as MLPs with intermediate linear layers followed by LayerNorm (Ba et al., 2016) and Mish (Misra, 2019) activations. To mitigate exploding gradients, we normalize the latent representation by projecting $\mathbf{z}$ into $L$ fixed-dimensional simplices using a softmax operation (Lavoie et al., 2022). A key benefit of embedding $\mathbf{z}$ as simplices (as opposed to *e.g.* a discrete representation or squashing) is that it naturally biases the representation towards sparsity without enforcing hard constraints (see Appendix H for motivation and implementation). We dub this normalization scheme *SimNorm*. Let $V$ be the dimensionality of each simplex $\mathbf{g}$ constructed from $L$ partitions (groups) of $\mathbf{z}$. SimNorm then applies the following transformation:

$$\mathbf{z}^\circ \doteq [\mathbf{g}_i, \ldots, \mathbf{g}_L], \ \mathbf{g}_i = \frac{e^{\mathbf{z}_{i:i+V}/\tau}}{\sum_{j=1}^{V} e^{\mathbf{z}_{i:i+V}/\tau}}, \quad (5)$$

where $\mathbf{z}^\circ$ is the simplicial embedding of $\mathbf{z}$, $[\cdot]$ denotes concatenation, and $\tau > 0$ is a temperature parameter that modulates the "sparsity" of the representation. As we will demonstrate in our experiments, SimNorm is essential to the training stability of TD-MPC**2**. Finally, to reduce bias in TD-targets generated by $\bar{Q}$, we learn an *ensemble* of $Q$-functions using the objective from Equation 3 and maintain $\bar{Q}$ as an EMA of each $Q$-function. We use 5 $Q$-functions in practice. Targets are then computed as the minimum of two randomly sub-sampled $\bar{Q}$-functions (Chen et al., 2021).

### 3.2 MODEL PREDICTIVE CONTROL WITH A POLICY PRIOR

TD-MPC**2** derives its closed-loop control policy by planning with the learned world model. Specifically, our approach leverages the MPC framework for local trajectory optimization using Model Predictive Path Integral (MPPI) (Williams et al., 2015) as a derivative-free optimizer with sampled action sequences $(\mathbf{a}_t, \mathbf{a}_{t+1}, \ldots, \mathbf{a}_{t+H})$ of length $H$ evaluated by rolling out *latent* trajectories with the model. At each decision step, we estimate parameters $\mu^*, \sigma^*$ of a time-dependent multivariate Gaussian with diagonal covariance such that expected return is maximized, *i.e.*,

$$\mu^*, \sigma^* = \arg\max_{(\mu, \sigma)} \mathbb{E}_{(\mathbf{a}_t, \mathbf{a}_{t+1}, \ldots, \mathbf{a}_{t+H}) \sim \mathcal{N}(\mu, \sigma^2)} \left[ \gamma^H Q(\mathbf{z}_{t+H}, \mathbf{a}_{t+H}) + \sum_{h=t}^{H-1} \gamma^h R(\mathbf{z}_h, \mathbf{a}_h) \right], \quad (6)$$

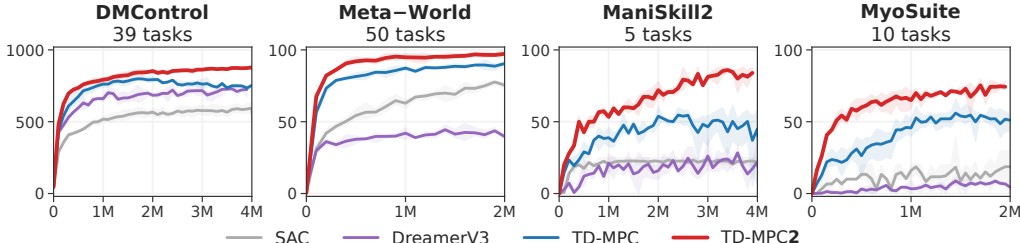

*Figure 4.* **Single-task RL.** Episode return (DMControl) and success rate (others) as a function of environment steps across **104** continuous control tasks spanning 4 diverse task domains. TD-MPC**2** achieves higher data-efficiency and asymptotic performance than existing methods, while using the **same** hyperparameters across all tasks. Mean and 95% CIs over 3 seeds.

where $\mu, \sigma \in \mathbb{R}^{H \times m}$, $\mathcal{A} \in \mathbb{R}^m$. Equation 6 is solved by iteratively sampling action sequences from $\mathcal{N}(\mu, \sigma^2)$, evaluating their expected return, and updating $\mu, \sigma$ based on a weighted average. Notably, Equation 6 estimates the full RL objective introduced in Section 2 by bootstrapping with the learned terminal value function beyond horizon $H$. TD-MPC**2** repeats this iterative planning process for a fixed number of iterations and executes the first action $\mathbf{a}_t \sim \mathcal{N}(\mu_t^*, \sigma_t^*)$ in the environment. To accelerate convergence of planning, a fraction of action sequences originate from the policy prior $p$, and we warm-start planning by initializing $(\mu, \sigma)$ as the solution to the previous decision step shifted by 1. Refer to Hansen et al. (2022) for more details about the planning procedure.

### 3.3 TRAINING GENERALIST TD-MPC**2** AGENTS

The success of TD-MPC**2** in diverse single-task problems can be attributed to the algorithm outlined above. However, learning a large generalist TD-MPC**2** agent that performs a variety of tasks across multiple task domains, embodiments, and action spaces poses several unique challenges: *(i)* how to learn and represent task semantics? *(ii)* how to accommodate multiple observation and action spaces without specific domain knowledge? *(iii)* how to leverage the learned model for few-shot learning of new tasks? We describe our approach to multitask model learning in the following.

**Learnable task embeddings.** To succeed in a multitask setting, an agent needs to learn a common representation that takes advantage of task similarities, while still retaining the ability to differentiate between tasks at test-time. When task or domain knowledge is available, *e.g.* in the form of natural language instructions, the task embedding $\mathbf{e}$ from Equation 2 may encode such information. However, in the general case where domain knowledge cannot be assumed, we may instead choose to *learn* the task embeddings (and, implicitly, task relations) from data. TD-MPC**2** conditions all of its five components with a learnable, fixed-dimensional task embedding $\mathbf{e}$, which is jointly trained together with other components of the model. To improve training stability, we constrain the $\ell_2$-norm of $\mathbf{e}$ to be $\leq 1$; this also leads to more semantically coherent task embeddings in our experiments. When finetuning a multitask TD-MPC**2** agent to a new task, we can choose to either initialize $\mathbf{e}$ as the embedding of a semantically similar task, or simply as a random vector.

**Action masking.** TD-MPC**2** learns to perform tasks with a variety of observation and action spaces, without any domain knowledge. To do so, we zero-pad all model inputs and outputs to their largest respective dimensions, and mask out invalid action dimensions in predictions made by the policy prior $p$ during both training and inference. This ensures that prediction errors in invalid dimensions do not influence TD-target estimation, and prevents $p$ from falsely inflating its entropy for tasks with small action spaces. We similarly only sample actions along valid dimensions during planning.

## 4 EXPERIMENTS

We evaluate TD-MPC**2** across a total of **104** diverse continuous control tasks spanning 4 task domains: DMControl (Tassa et al., 2018), Meta-World (Yu et al., 2019), ManiSkill2 (Gu et al., 2023), and MyoSuite (Caggiano et al., 2022). Tasks include high-dimensional state and action spaces (up to $\mathcal{A} \in \mathbb{R}^{39}$), sparse rewards, multi-object manipulation, physiologically accurate musculoskeletal motor control, complex locomotion (*e.g.* Dog and Humanoid embodiments), and cover a wide range of task difficulties. In support of open-source science, **we publicly release 300+ model checkpoints, datasets, and code for training and evaluating TD-MPC2 agents, which is available at https://tdmpc2.com**.

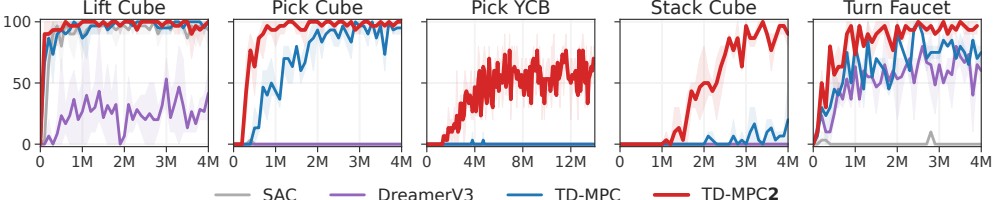

*Figure 5.* **High-dimensional locomotion.** Episode return as a function of environment steps in Humanoid ($\mathcal{A} \in \mathbb{R}^{21}$) and Dog ($\mathcal{A} \in \mathbb{R}^{38}$) locomotion tasks from DMControl. SAC and DreamerV3 are prone to numerical instabilities in Dog tasks, and are significantly less data-efficient than TD-MPC**2** in Humanoid tasks. Mean and $95\%$ CIs over 3 seeds. See Appendix D for more tasks.

*Figure 6.* **Object manipulation.** Success rate ($\%$) as a function of environment steps on 5 object manipulation tasks from ManiSkill2. *Pick YCB* considers manipulation of all 74 objects from the YCB (Calli et al., 2015) dataset. TD-MPC**2** excels at hard tasks. Mean and $95\%$ CIs over 3 seeds.

We seek to answer three core research questions through experimentation:

- **Comparison to existing methods.** How does TD-MPC**2** compare to state-of-the-art model-free (SAC) and model-based (DreamerV3, TD-MPC) methods for data-efficient continuous control?
- **Scaling.** Do the algorithmic innovations of TD-MPC**2** lead to improved agent capabilities as model and data size increases? Can a single agent learn to perform diverse skills across multiple task domains, embodiments, and action spaces?
- **Analysis.** How do the specific design choices introduced in TD-MPC**2** influence downstream task performance? How much does planning contribute to its success? Are the learned task embeddings semantically meaningful? Can large multi-task agents be adapted to unseen tasks?

**Baselines.** Our baselines represent the state-of-the-art in data-efficient RL, and include *(1)* **Soft Actor-Critic** (SAC) (Haarnoja et al., 2018), a model-free actor-critic algorithm based on maximum entropy RL, *(2)* **DreamerV3** (Hafner et al., 2023), a model-based method that optimizes a model-free policy with rollouts from a learned generative model of the environment, and *(3)* the original version of **TD-MPC** (Hansen et al., 2022), a model-based RL algorithm that performs local trajectory optimization (planning) in the latent space of a learned *implicit* (non-generative) world model. SAC and TD-MPC use task-specific hyperparameters, whereas TD-MPC**2** uses the **same** hyperparameters across all tasks. Additionally, it is worth noting that both SAC and TD-MPC use a larger batch size of 512, while 256 is sufficient for stable learning with TD-MPC**2**. Similarly, DreamerV3 uses a high update-to-data (UTD) ratio of 512, whereas TD-MPC**2** uses a UTD of 1 by default. We use a 5M parameter TD-MPC**2** agent in all experiments (unless stated otherwise). For reference, the DreamerV3 baseline has approx. 20M learnable parameters. See Appendix H for more details.

## 4.1 RESULTS

**Comparison to existing methods.** We first compare the data-efficiency of TD-MPC**2** to a set of strong baselines on **104** diverse tasks in an online RL setting. Aggregate results are shown in Figure 4. We find that TD-MPC**2** outperforms prior methods across all task domains. The MyoSuite results are particularly noteworthy, as we did not run *any* TD-MPC**2** experiments on this benchmark prior to the reported results. Individual task performances on some of the most difficult tasks (high-dimensional locomotion and multi-object manipulation) are shown in Figure 5 and Figure 6. TD-MPC**2** outperforms baselines by a large margin on these tasks, despite using the same hyperparameters across all tasks. Notably, TD-MPC sometimes diverges due to exploding gradients, whereas TD-MPC**2** remains stable. We provide per-task visualization of gradients in Appendix G. Similarly, we observe that DreamerV3 experiences occasional numerical instabilities (*Dog*) and

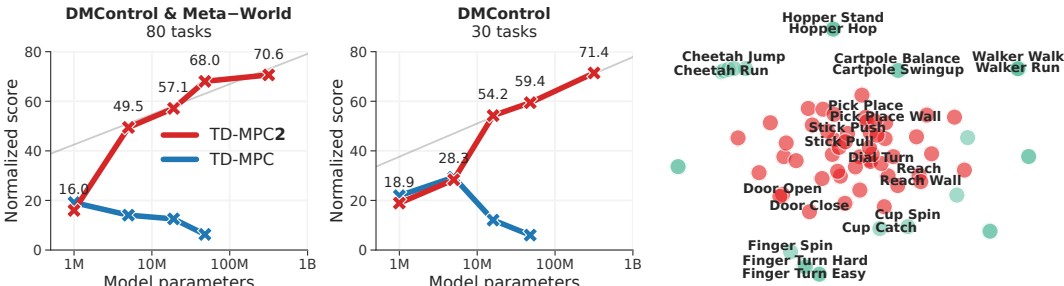

*Figure 7.* **Massively multi-task world models.** *(Left)* Normalized score as a function of model size on the two 80-task and 30-task datasets. TD-MPC**2** capabilities scale with model size. *(Right)* T-SNE (van der Maaten & Hinton, 2008) visualization of task embeddings learned by a TD-MPC**2** agent trained on 80 tasks from DMControl and Meta-World. A subset of labels are shown for clarity.

generally struggles with tasks that require fine-grained object manipulation (*lift*, *pick*, *stack*). See Appendix D for the full single-task RL results.

**Massively multitask world models.** To demonstrate that our proposed improvements facilitate scaling of world models, we evaluate the performance of 5 multitask models ranging from 1M to 317M parameters on a collection of **80** diverse tasks that span multiple task domains and vary greatly in objective, embodiment, and action space. Models are trained on a dataset of 545M transitions obtained from the replay buffers of 240 single-task TD-MPC**2** agents, and thus contain a wide variety of behaviors ranging from random to expert policies. The task set consists of all 50 Meta-World tasks, as well as 30 DMControl tasks. The DMControl task set includes 19 original DMControl tasks, as well as 11 new tasks. For completeness, we include a separate set of scaling results on the 30-task DMControl subset (345M transitions) as well. Due to our careful design of the TD-MPC**2** algorithm, scaling up is straightforward:

*Table 1.* **Training cost.** Approximate TD-MPC**2** training cost on the 80-task dataset, reported in GPU days on a single NVIDIA GeForce RTX 3090 GPU. We also list the normalized score achieved by each model at end of training.

| Params (M) | GPU days | Score |
|---|---|---|
| 1 | 3.7 | 16.0 |
| 5 | 4.2 | 49.5 |
| 19 | 5.3 | 57.1 |
| 48 | 12 | 68.0 |
| **317** | **33** | **70.6** |

to improve rate of convergence we use a $4\times$ larger batch size (1024) compared to the single-task experiments, but make no other changes to hyperparameters.

**Scaling TD-MPC2 to 317M parameters.** Our scaling results are shown in Figure 7. To summarize agent performance with a single metric, we produce a normalized score that is an average of all individual task success rates (Meta-World) and episode returns normalized to the [0, 100] range (DMControl). We observe that agent capabilities consistently increase with model size on both task sets. Notably, performance does not appear to have saturated for our largest models (317M parameters) on either dataset, and we can thus expect results to continue improving beyond our considered model sizes. We refrain from formulating a scaling law, but note that normalized score appears to scale linearly with the log of model parameters (gray line in Figure 7). We also report approximate training costs in Table 1. The 317M parameter model can be trained with limited computational resources. To better understand why multitask model learning is successful, we explore the task embeddings learned by TD-MPC**2** (Figure 7, right). Intriguingly, tasks that are semantically similar (*e.g.*, Door Open and Door Close) are close in the learned task embedding space. However,

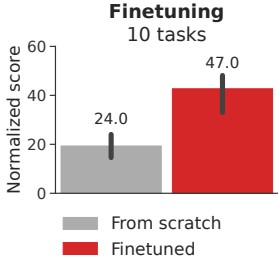

*Figure 8.* **Finetuning.** Score of a 19M parameter TD-MPC**2** agent trained on 70 tasks and finetuned online to each of 10 held-out tasks for 20k environment steps. 3 seeds.

embedding similarity appears to align more closely with task *dynamics* (embodiment, objects) than objective (walk, run). This makes intuitive sense, as dynamics are tightly coupled with control.

**Few-shot learning.** While our work mainly focuses on the *scaling* and *robustness* of world models, we also explore the efficacy of finetuning pretrained world models for few-shot learning of unseen tasks. Specifically, we pretrain a 19M parameter TD-MPC**2** agent on 70 tasks from DMControl and Meta-World, and naïvely finetune the full model to each of 10 held-out tasks (5 from each domain) via online RL with an initially empty replay buffer and no changes to hyperparameters. Aggregate

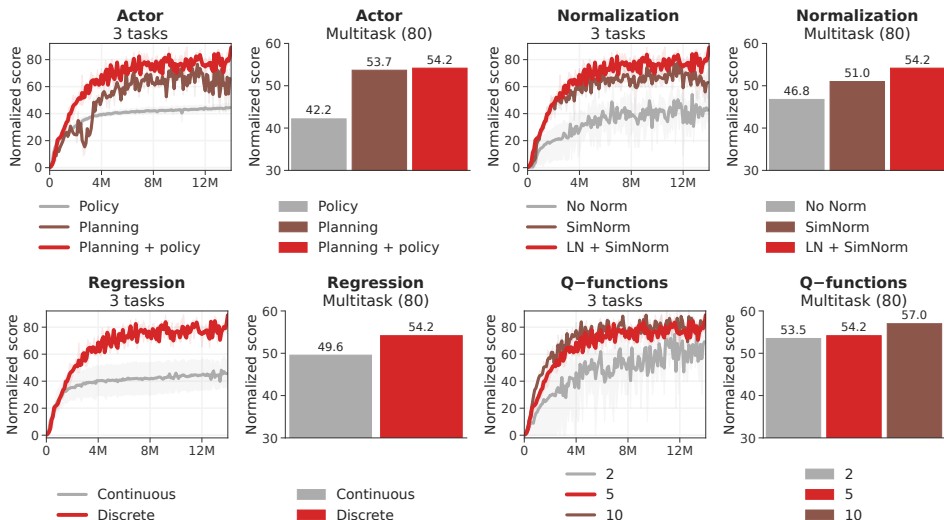

*Figure 9.* **Ablations.** *(Curves)* Normalized score as a function of environment steps, averaged across three of the most difficult tasks: *Dog Run*, *Humanoid Walk* (DMControl), and *Pick YCB* (ManiSkill2). Mean and $95\%$ CIs over 3 random seeds. *(Bars)* Normalized score of 19M parameter multitask (80 tasks) TD-MPC**2** agents. Our ablations highlight the relative importance of each design choice; **red** is the default formulation of TD-MPC**2**. See Appendix D for more ablations.

results are shown in Figure 8. We find that TD-MPC**2** improves $2\times$ over learning from scratch on new tasks in the low-data regime (20k environment steps[1]). Although finetuning world models to new tasks is very much an open research problem, our exploratory results are promising. See Appendix E for experiment details and individual task curves.

**Ablations.** We ablate most of our design choices for TD-MPC**2**, including choice of actor, various normalization techniques, regression objective, and number of $Q$-functions. Our main ablations, shown in Figure 9, are conducted on three of the most difficult online RL tasks, as well as large-scale multitask training (80 tasks). We observe that all of our proposed improvements contribute meaningfully to the robustness and strong performance of TD-MPC**2** in both single-task RL and multi-task RL. Interestingly, we find that the relative importance of each design choice is consistent across both settings. Lastly, we also ablate normalization of the learned task embeddings, shown in Appendix F. The results indicate that maintaining a normalized task embedding space ($\ell_2$-norm of 1) is moderately important for stable multitask training, and results in more meaningful task relations.

## 5 LESSONS, OPPORTUNITIES, AND RISKS

**Lessons.** Historically, RL algorithms have been notoriously sensitive to architecture, hyperparameters, characteristics of the task, and even random seed (Henderson et al., 2018), with no principled method for tuning the algorithms. As a result, successful application of deep RL often requires large teams of experts with significant computational resources (Berner et al., 2019; Schrittwieser et al., 2020; Ouyang et al., 2022). TD-MPC**2** – along with several other contemporary RL methods (Yarats et al., 2021; Ye et al., 2021; Hafner et al., 2023) – seek to democratize use of RL (*i.e.*, lowering the barrier of entry for smaller teams of academics, practitioners, and individuals with fewer resources) by improving robustness of existing open-source algorithms. **We firmly believe that improving algorithmic robustness will continue to have profound impact on the field.** A key lesson from the development of TD-MPC**2** is that the community has yet to discover an algorithm that truly masters *everything* out-of-the-box. While *e.g.* DreamerV3 (Hafner et al., 2023) has delivered strong results on challenging tasks with discrete action spaces (such as Atari games and Minecraft), we find that TD-MPC**2** produces significantly better results on difficult continuous control tasks. At the same time, extending TD-MPC**2** to discrete action spaces remains an open problem.

**Opportunities.** Our scaling results demonstrate a path for model-based RL in which massively multitask world models are leveraged as *generalist* world models. While multi-task world models

---

[1]20k environment steps corresponds to 20 episodes in DMControl and 100 episodes in Meta-World.

remain relatively underexplored in literature, prior work suggests that the implicit world model of TD-MPC**2** may be better suited than reconstruction-based approaches for tasks with large visual variation (Zhu et al., 2023). We envision a future in which implicit world models are used zero-shot to perform diverse tasks on *seen* embodiments (Xu et al., 2023; Yang et al., 2023), finetuned to quickly perform tasks on *new* embodiments, and combined with existing vision-language models to perform higher-level cognitive tasks in conjunction with low-level physical interaction. Our results are promising, but such level of generalization will likely require several orders of magnitude more tasks than currently available. Lastly, we want to remark that, while TD-MPC**2** relies on rewards for task learning, it is useful to adopt a generalized notion of reward as simply a metric for task completion. Such metrics already exist in the wild, *e.g.*, success labels, human preferences or interventions (Ouyang et al., 2022), or the embedding distance between a current observation and a goal (Eysenbach et al., 2022; Ma et al., 2022) within a pre-existing learned representation. However, leveraging such rewards for large-scale pretraining is an open problem. To accelerate research in this area, we are releasing **300+** TD-MPC**2** models, including 12 multitask models, as well as datasets and code, and we are beyond excited to see what the community will do with these resources.

**Risks.** While we are excited by the potential of generalist world models, several challenges remain: *(i)* misspecification of task rewards can lead to unintended outcomes (Clark & Amodei, 2016) that may be difficult to anticipate, *(ii)* handing over unconstrained autonomy of physical robots to a learned model can result in catastrophic failures if no additional safety checks are in place (Lancaster et al., 2023), and *(iii)* data for certain applications may be prohibitively expensive for small teams to obtain at the scale required for generalist behavior to emerge, leading to a concentration of power. Mitigating each of these challenges will require new research innovations, and we invite the community to join us in these efforts.

## 6 RELATED WORK

Multiple prior works have sought to build RL algorithms that are robust to hyperparameters, architecture, as well as variation in tasks and data. For example, *(1)* Double $Q$-learning (Hasselt et al., 2016), RED-Q (Chen et al., 2021), SVEA (Hansen et al., 2021), and SR-SPR (D'Oro et al., 2023) each improve the stability of $Q$-learning algorithms by adjusting the bias-variance trade-off in TD-target estimation, *(2)* C51 (Bellemare et al., 2017) and DreamerV3 (Hafner et al., 2023) improve robustness to the magnitude of rewards by performing discrete regression in a transformed space, and *(3)* model-free algorithms DrQ (Kostrikov et al., 2020) and DrQ-v2 (Yarats et al., 2021) improve training stability and exploration, respectively, through use of data augmentation and several other minor but important implementation details. However, all of the aforementioned works strictly focus on improving data-efficiency and robustness in single-task online RL.

Existing literature that studies scaling of neural architectures for decision-making typically assume access to large datasets of near-expert demonstrations for behavior cloning (Reed et al., 2022; Lee et al., 2022; Kumar et al., 2022; Schubert et al., 2023; Driess et al., 2023; Brohan et al., 2023). Gato (Reed et al., 2022) learns to perform tasks across multiple domains by training a large Transformer-based sequence model (Vaswani et al., 2017) on an enormous dataset of expert demonstrations, and RT-1 (Brohan et al., 2023) similarly learns a sequence model for object manipulation on a single (real) robot embodiment by training on a large dataset collected by human teleoperation. While the empirical results of this line of work are impressive, the assumption of large demonstration datasets is impractical. Additionally, current sequence models rely on discretization of the action space (tokenization), which makes scaling to high-dimensional continuous control tasks difficult.

Most recently, researchers have explored scaling of RL algorithms as a solution to the aforementioned challenges (Baker et al., 2022; Jia et al., 2022; Xu et al., 2023; Kumar et al., 2023; Hafner et al., 2023). For example, VPT (Baker et al., 2022) learns to play Minecraft by first pretraining a behavior cloning policy on a large human play dataset, and then finetuning the policy with RL. GSL (Jia et al., 2022) requires no pre-existing data. Instead, GSL iteratively trains a population of "specialist" agents on individual task variations, distills them into a "generalist" policy via behavior cloning, and then uses the generalist as initialization for the next population of specialists. However, this work considers strictly single-task RL and assumes full control over the initial state in each episode. Lastly, DreamerV3 (Hafner et al., 2023) successfully scales its world model in terms of parameters and shows that larger models generally are more data-efficient in an online RL setting, but does not consider multitask RL.

ACKNOWLEDGEMENTS

This project was supported, in part, by grants from NSF CAREER Award (2240160), NSF TILOS AI Institute (2112665), NSF CCF-2112665 (TILOS), NSF 1730158 CI-New: Cognitive Hardware and Software Ecosystem Community Infrastructure (CHASE-CI), NSF ACI-1541349 CC*DNI Pacific Research Platform, and gifts from Qualcomm.

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

APPENDICES

## A SUMMARY OF IMPROVEMENTS

We summarize the main differences between TD-MPC and TD-MPC2 as follows:

- **Architectural design.** All components of TD-MPC2 are MLPs with LayerNorm (Ba et al., 2016) and Mish (Misra, 2019) activations after each layer. We apply SimNorm normalization to the latent state $\mathbf{z}$ which biases the representation towards sparsity and maintaining a small $\ell_2$-norm. We train an ensemble of $Q$-functions (5 by default) and additionally apply 1% Dropout (Srivastava et al., 2014) after the first linear layer in each $Q$-function. TD-targets are computed as the minimum of two randomly subsampled $Q$-functions (Chen et al., 2021). In contrast, TD-MPC is implemented as MLPs without LayerNorm, and instead uses ELU (Clevert et al., 2015) activations. TD-MPC does not constrain the latent state at all, which in some instances leads to exploding gradients (see Appendix G for experimental results). Lastly, TD-MPC learns only 2 $Q$-functions and does not use Dropout. The architectural differences in TD-MPC2 result in a 4M net increase in learnable parameters (5M total) for our default single-task model size compared to the 1M parameters of TD-MPC. However, as shown in Figure 7, naïvely increasing the model size of TD-MPC does not lead to consistently better performance, whereas it does for TD-MPC2.

- **Policy prior.** The policy prior of TD-MPC2 is trained with maximum entropy RL (Ziebart et al., 2008; Haarnoja et al., 2018), whereas the policy prior of TD-MPC is trained as a deterministic policy with Gaussian noise applied to actions. We find that a carefully tuned Gaussian noise schedule is comparable to a policy prior trained with maximum entropy. However, maximum entropy RL can more easily be applied with task-agnostic hyperparameters. We only compute policy entropy over valid action dimensions in multi-task learning with multiple action spaces.

- **Planning.** The planning procedure of TD-MPC2 closely follows that of TD-MPC. However, we simplify planning marginally by not leveraging momentum between iteration, as we find it to produce comparable results. We also improve the throughput of planning by approx. $\mathbf{2}\times$ through a series of code-level optimizations.

- **Model objective.** We revisit the training objective of TD-MPC and improve its robustness to variation in tasks, such as the magnitude of rewards. TD-MPC2 uses discrete regression (soft cross-entropy) of rewards and values in a $\log$-transformed space, which makes the magnitude of the two loss terms independent of the magnitude of the task rewards. TD-MPC uses continuous regression which leads to training instabilities in tasks where rewards are large. While this issue can be alleviated by, *e.g.*, normalizing task rewards based on moving statistics, in the single-task case, it is difficult to design robust reward normalization schemes for multi-task learning. TD-MPC2 retains the continuous regression term for joint-embedding prediction as the latent representation is already normalized by SimNorm, and discrete regression is computationally expensive for high-dimensional spaces (requires $N$ bins for each dimension of $\mathbf{z}$).

- **Multi-task model.** TD-MPC2 introduces a framework for learning multi-task world models across multiple domains, embodiments, and action spaces. We introduce a normalized learnable task embedding space which all components of TD-MPC are conditioned on, and we accommodate multiple observation and action spaces by applying zero-padding and action masking during both training and inference. We train multi-task models on a large number of tasks, and finetune the model to held-out tasks (across embodiments) using online RL. TD-MPC only considers multi-task learning on a small number of tasks with shared observation and action space, and does not consider finetuning of the learned multi-task model.

- **Simplified algorithm and implementation.** TD-MPC2 removes momentum in MPPI (Williams et al., 2015), and replaces prioritized experience replay sampling from the replay buffer with uniform sampling, both of which simplify the implementation with no significant change in experimental results. Finally, we also use a faster replay buffer implementation that uses multiple workers for sampling, and we increase training and planning throughput through code-level optimizations such as $Q$-function ensemble vectorization, which makes the wall-time of TD-MPC2 comparable to that of TD-MPC despite a larger architecture (5M vs. 1M).

# B  TASK VISUALIZATIONS

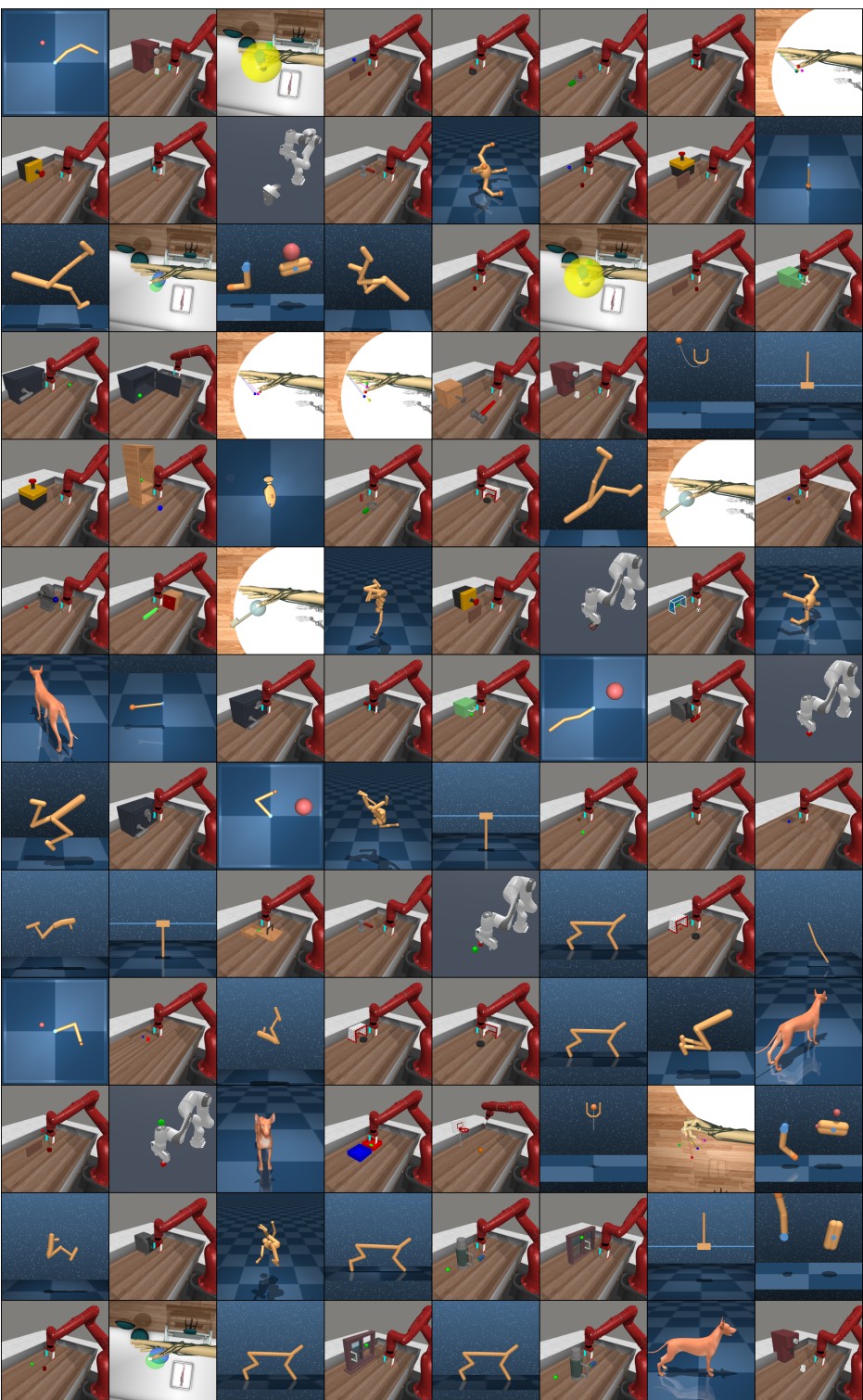

*Figure 10.* **Task visualizations.** Visualization of a random initial state for each of the **104** tasks that we consider. Tasks vary greatly in objective, embodiment, and action space. Visit https://tdmpc2.com for videos of TD-MPC**2** performing each task. See Appendix C for task details.

# C  TASK DOMAINS

We consider a total of 104 continuous control tasks from 4 task domains: DMControl (Tassa et al., 2018), Meta-World (Yu et al., 2019), ManiSkill2 (Gu et al., 2023), and MyoSuite (Caggiano et al., 2022). This section provides an exhaustive list of all tasks considered, as well as their observation and action dimensions. Environment details are listed at the end of the section. We provide (static) task visualizations in Appendix B and videos of TD-MPC**2** agents performing each task at `https://www.tdmpc2.com`.

*Table 2.* **DMControl.** We consider a total of 39 continuous control tasks in the DMControl domain, including 19 original DMControl tasks and 11 new (custom) tasks created specifically for TD-MPC**2** benchmarking and multitask training. We list all considered DMControl tasks below. The *Locomotion* task set shown in Figure 1 corresponds to the *Humanoid* and *Dog* embodiments of DMControl, with performance reported at 14M environment steps.

| Task | Observation dim | Action dim | Sparse? | New? |
| --- | --- | --- | --- | --- |
| Acrobot Swingup | 6 | 1 | N | N |
| Cartpole Balance | 5 | 1 | N | N |
| Cartpole Balance Sparse | 5 | 1 | Y | N |
| Cartpole Swingup | 5 | 1 | N | N |
| Cartpole Swingup Sparse | 5 | 1 | Y | N |
| Cheetah Jump | 17 | 6 | N | Y |
| Cheetah Run | 17 | 6 | N | N |
| Cheetah Run Back | 17 | 6 | N | Y |
| Cheetah Run Backwards | 17 | 6 | N | Y |
| Cheetah Run Front | 17 | 6 | N | Y |
| Cup Catch | 8 | 2 | Y | N |
| Cup Spin | 8 | 2 | N | Y |
| Dog Run | 223 | 38 | N | N |
| Dog Trot | 223 | 38 | N | N |
| Dog Stand | 223 | 38 | N | N |
| Dog Walk | 223 | 38 | N | N |
| Finger Spin | 9 | 2 | Y | N |
| Finger Turn Easy | 12 | 2 | Y | N |
| Finger Turn Hard | 12 | 2 | Y | N |
| Fish Swim | 24 | 5 | N | N |
| Hopper Hop | 15 | 4 | N | N |
| Hopper Hop Backwards | 15 | 4 | N | Y |
| Hopper Stand | 15 | 4 | N | N |
| Humanoid Run | 67 | 24 | N | N |
| Humanoid Stand | 67 | 24 | N | N |
| Humanoid Walk | 67 | 24 | N | N |
| Pendulum Spin | 3 | 1 | N | Y |
| Pendulum Swingup | 3 | 1 | N | N |
| Quadruped Run | 78 | 12 | N | N |
| Quadruped Walk | 78 | 12 | N | N |
| Reacher Easy | 6 | 2 | Y | N |
| Reacher Hard | 6 | 2 | Y | N |
| Reacher Three Easy | 8 | 3 | Y | Y |
| Reacher Three Hard | 8 | 3 | Y | Y |
| Walker Run | 24 | 6 | N | N |
| Walker Run Backwards | 24 | 6 | N | Y |
| Walker Stand | 24 | 6 | N | N |
| Walker Walk | 24 | 6 | N | N |
| Walker Walk Backwards | 24 | 6 | N | Y |

*Table 3.* **Meta-World.** We consider a total of 50 continuous control tasks from the Meta-World domain. The Meta-World benchmark is designed for multitask and meta-learning research and all tasks thus share embodiment, observation space, and action space.

| Task | Observation dim | Action dim |
|------|-----------------|------------|
| Assembly | 39 | 4 |
| Basketball | 39 | 4 |
| Bin Picking | 39 | 4 |
| ... | ... | ... |
| Window Open | 39 | 4 |

*Table 4.* **ManiSkill2.** We consider a total of 5 continuous control tasks from the ManiSkill2 domain. The ManiSkill2 benchmark is designed for large-scale robot learning and contains a high degree of randomization and task variations. The *Pick YCB* task shown in Figure 1 corresponds to the ManiSkill2 task of the same name, with performance reported at 14M environment steps.

| Task | Observation dim | Action dim |
|------|-----------------|------------|
| Lift Cube | 42 | 4 |
| Pick Cube | 51 | 4 |
| Pick YCB | 51 | 7 |
| Stack Cube | 55 | 4 |
| Turn Faucet | 40 | 7 |

*Table 5.* **MyoSuite.** We consider a total of 10 continuous control tasks from the MyoSuite domain. The MyoSuite benchmark is designed for high-dimensional physiologically accurate muscoloskeletal motor control and involves particularly complex object manipulation with a dexterous hand. The MyoSuite domain consists of tasks with and without goal randomization. We consider both settings, and refer to them as *Easy* (fixed goal) and *Hard* (random goal), respectively.

| Task | Observation dim | Action dim |
|------|-----------------|------------|
| Reach Easy | 115 | 39 |
| Reach Hard | 115 | 39 |
| Pose Easy | 108 | 39 |
| Pose Hard | 108 | 39 |
| Pen Twirl Easy | 83 | 39 |
| Pen Twirl Hard | 83 | 39 |
| Object Hold Easy | 91 | 39 |
| Object Hold Hard | 91 | 39 |
| Key Turn Easy | 93 | 39 |
| Key Turn Hard | 93 | 39 |

**Environment details.** We benchmark algorithms on DMControl, Meta-World, ManiSkill2, and MyoSuite without modification. All four domains are infinite-horizon continuous control environments for which we use a fixed episode length and no termination conditions. We list episode lengths, action repeats, total number of environment steps, and the performance metric used for each domain in Table 6. In all experiments, we only consider an episode successful if the final step of an episode is successful. This is a stricter definition of success than used in some of the related literature, which *e.g.* may consider an episode successful if success is achieved at *any* step within a given episode. In tasks that require manipulation of objects, such as picking up an object, our definition of success ensures that an episode in which an object is picked up but then dropped again is not considered successful.

*Table 6.* **Environment details.** We list the episode length and action repeat used for each task domain, as well as the total number of environment steps and performance metrics that we use for benchmarking methods. All methods use the same values for all tasks.

|  | **DMControl** | **Meta-World** | **ManiSkill2** | **MyoSuite** |
|---|---|---|---|---|
| Episode length | $1,000$ | 200 | 200 | 100 |
| Action repeat | 2 | 2 | 2 | 1 |
| Effective length | 500 | 100 | 100 | 100 |
| Total env. steps | 4M - 14M | 2M | 4M - 14M | 2M |
| Performance metric | Reward | Success | Success | Success |

— **Appendices continue on next page** —

# D  SINGLE-TASK EXPERIMENTAL RESULTS

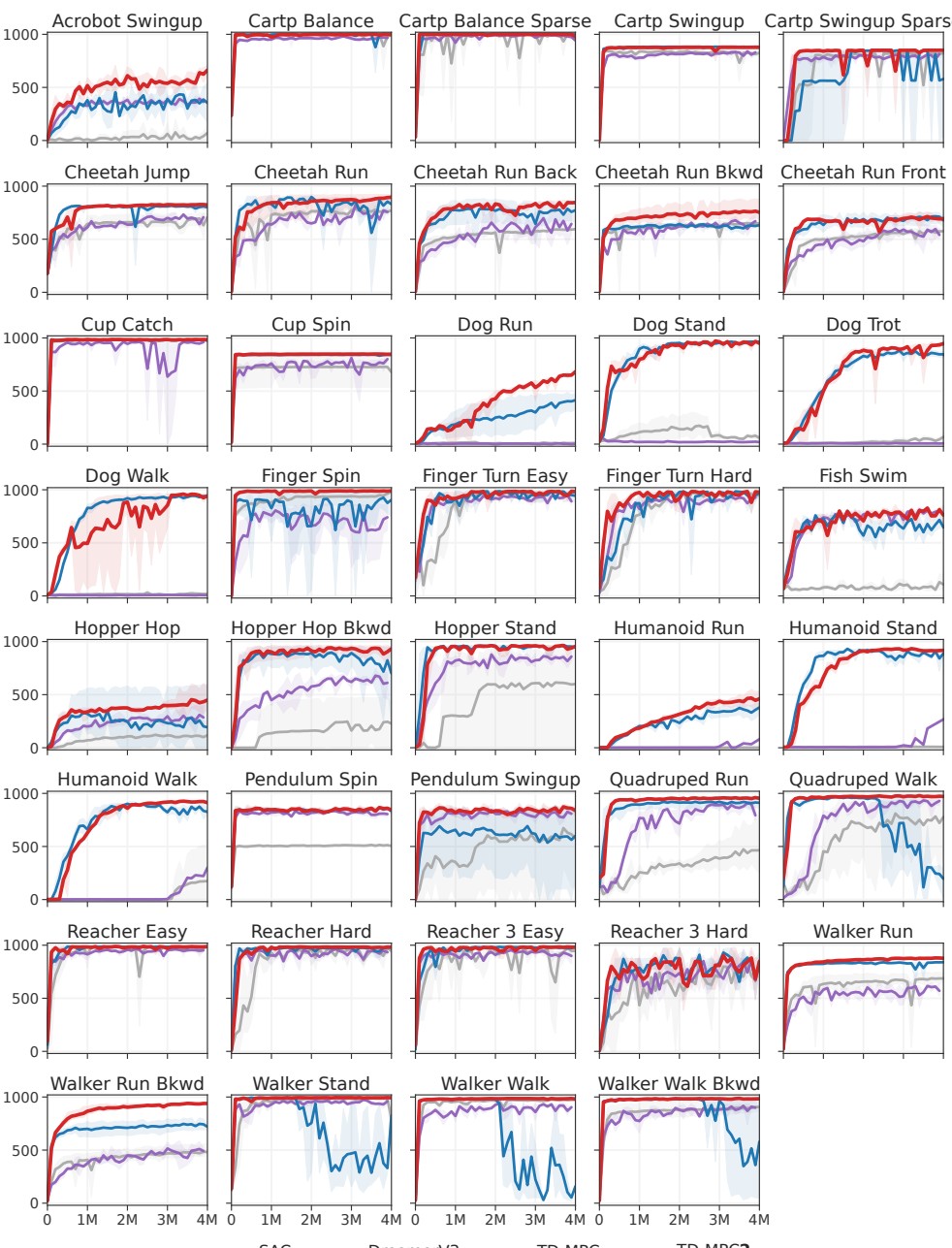

*Figure 11.* **Single-task DMControl results.** Episode return as a function of environment steps. The first 4M environment steps are shown for each task, although the Humanoid and Dog tasks are run for 14M environment steps; we provide those curves in Figure 14 as part of the "Locomotion" benchmark. Note that TD-MPC diverges on tasks like *Walker Stand* and *Walker Walk* whereas TD-MPC**2** remains stable. We visualize gradients on these tasks in Appendix G. Mean and 95% CIs over 3 seeds.

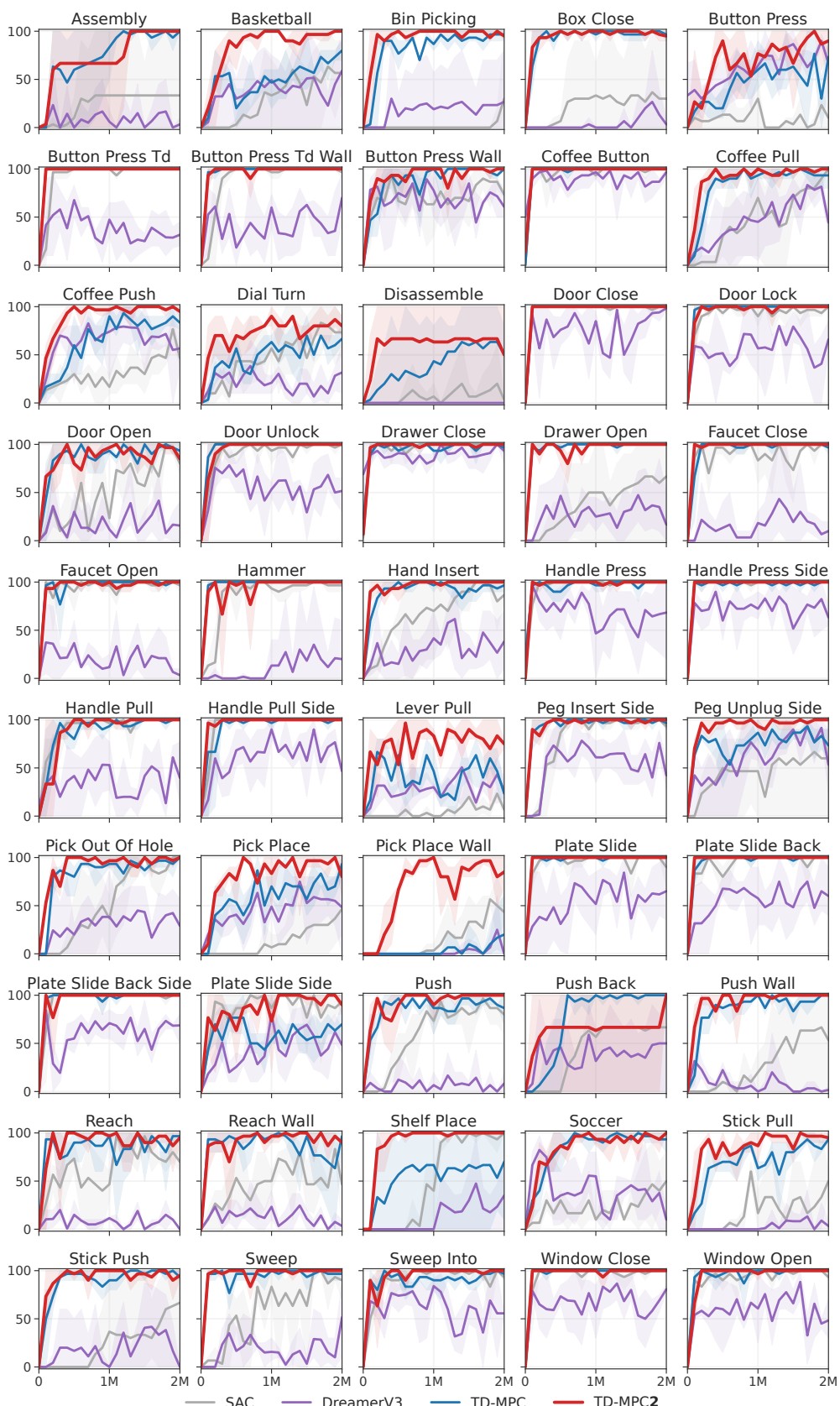

*Figure 12.* **Single-task Meta-World results.** Success rate (%) as a function of environment steps. TD-MPC**2** performance is comparable to existing methods on easy tasks, while outperforming other methods on hard tasks such as *Pick Place Wall* and *Shelf Place*. DreamerV3 often fails to converge.

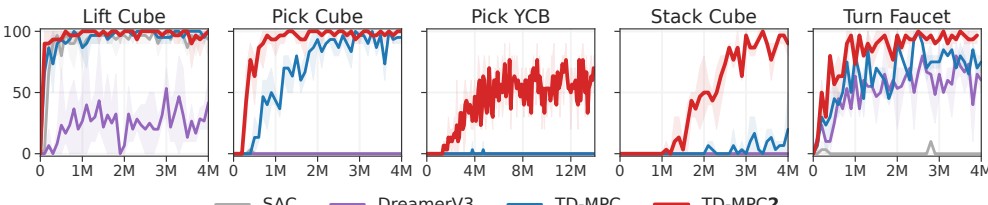

*Figure 13.* **Single-task ManiSkill2 results.** Success rate (%) as a function of environment steps on 5 object manipulation tasks from ManiSkill2. *Pick YCB* is the hardest task and considers manipulation of all 74 objects from the YCB (Calli et al., 2015) dataset. We report results for this tasks at 14M environment steps, and 4M environment steps for other tasks. TD-MPC**2** achieves a $> 60\%$ success rate on the Pick YCB task, whereas other methods fail to learn within the given budget. Mean and $95\%$ CIs over 3 seeds.

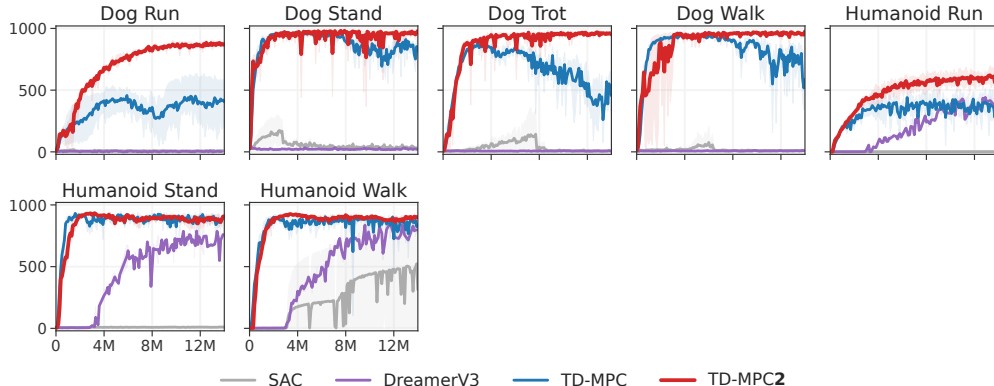

*Figure 14.* **Single-task high-dimensional locomotion results.** Episode return as a function of environment steps on all 7 "Locomotion" benchmark tasks. This domain includes high-dimensional Humanoid ($\mathcal{A} \in \mathbb{R}^{21}$) and Dog ($\mathcal{A} \in \mathbb{R}^{38}$) embodiments. Mean and $95\%$ CIs over 3 seeds.

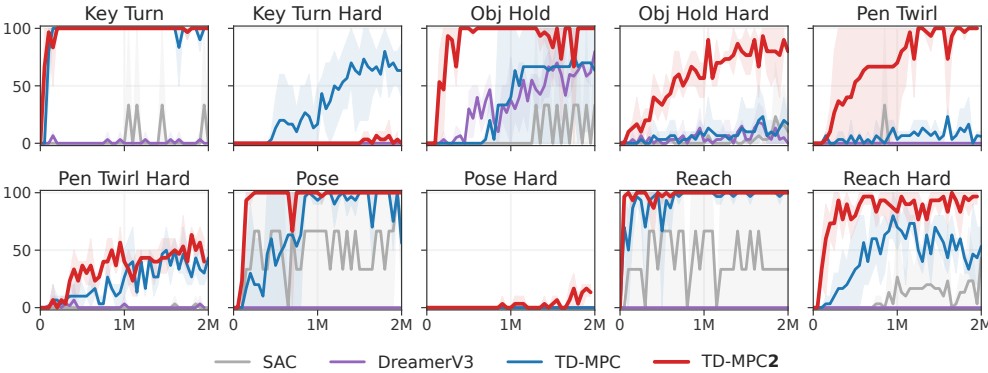

*Figure 15.* **Single-task MyoSuite results.** Success rate (%) as a function of environment steps. This task domain includes high-dimensional contact-rich musculoskeletal motor control ($\mathcal{A} \in \mathbb{R}^{39}$) with a physiologically accurate robot hand. Goals are randomized in tasks designated as "Hard". TD-MPC**2** achieves comparable or better performance than existing methods on all tasks from this benchmark, except for *Key Turn Hard* in which TD-MPC succeeds early in training.

## E FEW-SHOT EXPERIMENTAL RESULTS

We finetune a 19M parameter TD-MPC**2** agent trained on 70 tasks to each of 10 held-out tasks. Individual task curves are shown in Figure 16. We compare data-efficiency of the finetuned model to a baseline agent of similar model capacity trained from scratch. However, we find that performance of our 19M parameter baselines trained from scratch are comparable to our 5M parameter agents also trained from scratch. Our few-shot finetuning results suggest that the efficacy of finetuning is somewhat task-dependent. However, more research is needed to conclude whether this is due to task similarity (or rather lack thereof) or due to subpar task performance of the pretrained agent on the source task. We conjecture that both likely influence results.

When finetuning to an unseen task, we initialize the learnable task embedding for the new task as the embedding of a semantically similar task from the pretraining dataset. We list the source task embedding used as initialization for each experiment in Table 7. We did not experiment with other initialization schemes, nor other task pairings.

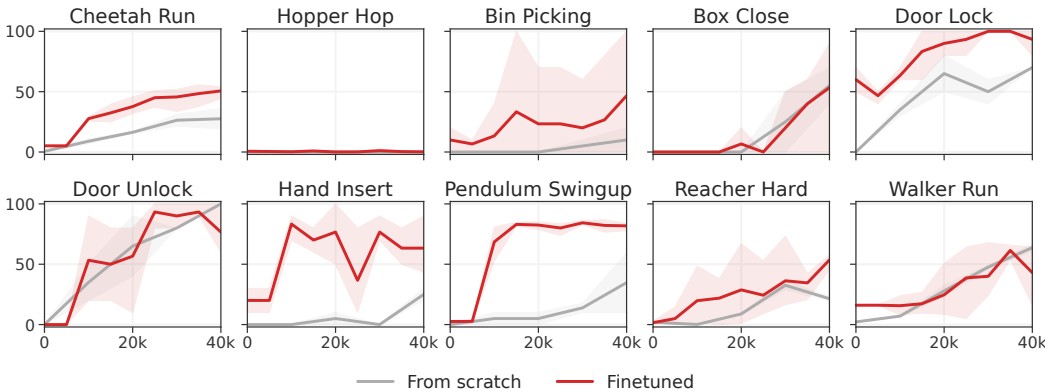

*Figure 16.* **Few-shot learning.** Normalized episode return (DMControl) and success rate (Meta-World) as a function of environment steps while finetuning a 19M parameter TD-MPC**2** agent trained on 70 tasks to each of 10 held-out tasks. 40k steps corresponds to 40 episodes in DM-Control and 200 in Meta-World. Mean and 95% CIs over 3 seeds.

*Table 7.* **Initialization of task embeddings for few-shot learning.** We list the task embeddings used as initialization when finetuning our 19M parameter TD-MPC**2** agent to held-out tasks. We did not experiment with other initialization schemes, nor other task pairings.

| Target task | Source task |
|---|---|
| Walker Run | Walker Walk |
| Cheetah Run | Cheetah Run Backwards |
| Hopper Hop | Hopper Stand |
| Pendulum Swingup | Pendulum Spin |
| Reacher Hard | Reacher Easy |
| Bin Picking | Pick Place |
| Box Close | Assembly |
| Door Lock | Door Open |
| Door Unlock | Door Open |
| Hand Insert | Sweep Into |

# F    ADDITIONAL ABLATIONS

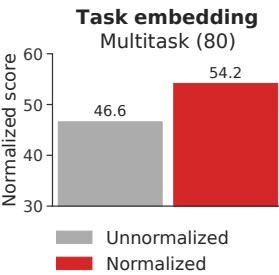

*Figure 17.* **Normalized task embeddings.** Normalized score of 19M parameter multitask (80 tasks) TD-MPC**2** agents, with and without normalized task embeddings **e** as described in Section 3.1. We find that normalizing **e** to have a maximum $\ell_2$-norm of 1 improves multitask performance.

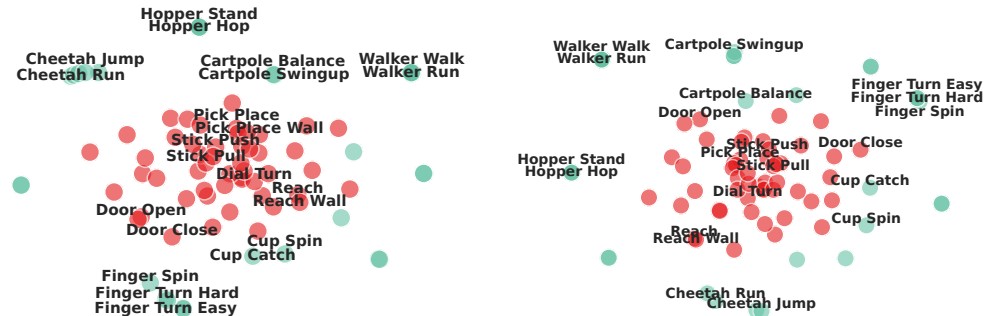

*Figure 18.* **T-SNE of task embeddings with and without normalization.** T-SNE (van der Maaten & Hinton, 2008) visualizations of task embeddings learned by TD-MPC**2** agent trained on 80 tasks from DMControl and Meta-World. *(Left) with* normalization. *(Right) without* normalization. A subset of labels are shown for clarity. We observe that task embeddings are more semantically meaningful when normalized during training, *e.g.*, "Door Open" and "Door Close" are close in embedding space on the left, but far apart on the right.

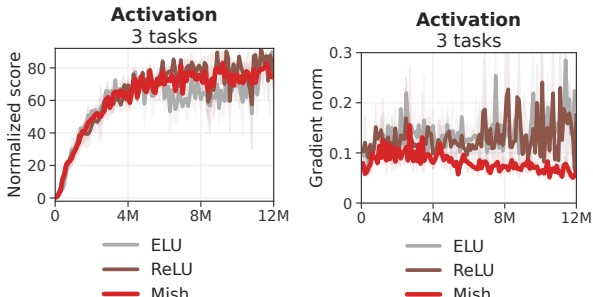

*Figure 19.* **Activation function.** Normalized score as a function of environment steps, averaged across three of the most difficult tasks: *Dog Run*, *Humanoid Walk* (DMControl), and *Pick YCB* (ManiSkill2). Mean and 95% CIs over 3 random seeds. We find that TD-MPC2 achieves comparable asymptotic performance and data-efficiency with either activation function, but that Mish (Misra, 2019) leads to smoother gradients overall.

## G  GRADIENT NORM AND TRAINING STABILITY

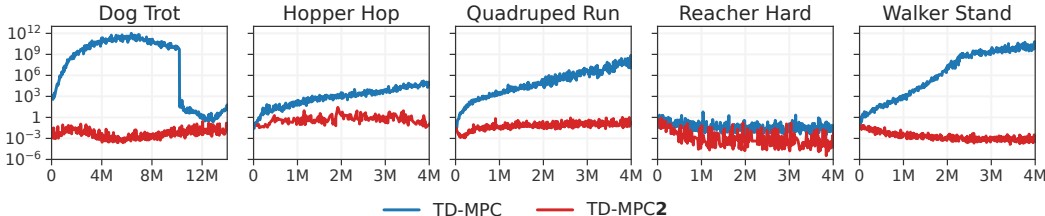

*Figure 20.* **Gradient norm during training.** We compare the gradient norm (log-scale) of TD-MPC and TD-MPC**2** as a function of environment steps on five tasks from DMControl. TD-MPC is prone to exploding gradients, which can cause learning to diverge on some tasks (*e.g.*, Walker Stand in Figure 11). In comparison, the gradients of TD-MPC**2** remain stable throughout training. We only display 1 seed per task for visual clarity.

## H  IMPLEMENTATION DETAILS

**Architectural details.** All components of TD-MPC**2** are implemented as MLPs. The encoder $h$ contains a variable number of layers $(2-5)$ depending on the architecture size; all other components are 3-layer MLPs. Intermediate layers consist of a linear layer followed by LayerNorm and a Mish activation function. The latent representation is normalized as a simplicial embedding. $Q$-functions additionally use Dropout. We summarize the TD-MPC**2** architecture for the 5M parameter `base` (default for online RL) model size using PyTorch-like notation:

```
Encoder parameters: 167,936
Dynamics parameters: 843,264
Reward parameters: 631,397
Policy parameters: 582,668
Q parameters: 3,156,985
Task parameters: 7,680
Total parameters: 5,389,930

Architecture: TD-MPC2 base 5M(
  (task_embedding): Embedding(T, 96, max_norm=1)
  (encoder): ModuleDict(
    (state): Sequential(
      (0): NormedLinear(in_features=S+T, out_features=256, act=Mish)
      (1): NormedLinear(in_features=256, out_features=512, act=SimNorm)
    )
  )
  (dynamics): Sequential(
    (0): NormedLinear(in_features=512+T+A, out_features=512, act=Mish)
    (1): NormedLinear(in_features=512, out_features=512, act=Mish)
    (2): NormedLinear(in_features=512, out_features=512, act=SimNorm)
  )
  (reward): Sequential(
    (0): NormedLinear(in_features=512+T+A, out_features=512, act=Mish)
    (1): NormedLinear(in_features=512, out_features=512, act=Mish)
    (2): Linear(in_features=512, out_features=101,)
  )
  (pi): Sequential(
    (0): NormedLinear(in_features=512+T, out_features=512, act=Mish)
    (1): NormedLinear(in_features=512, out_features=512, act=Mish)
    (2): Linear(in_features=512, out_features=2A, bias=True)
  )
  (Qs): Vectorized ModuleList(
    (0-4): 5 x Sequential(
      (0): NormedLinear(in_features=512+T+A, out_features=512, dropout=0.01, act=Mish)
      (1): NormedLinear(in_features=512, out_features=512, act=Mish)
      (2): Linear(in_features=512, out_features=101, bias=True)
    )
  )
)
```

where `S` is the input dimensionality, `T` is the number of tasks, and `A` is the action space. We exclude the task embedding `T` from single-task experiments. The exact parameter counts listed above are for `S`= 39, `T`= 80, and `A`= 6.

**Hyperparameters.** We use the same hyperparameters across all tasks. Our hyperparameters are listed in Table 8. We use the same hyperparameters for TD-MPC and SAC as in Hansen et al. (2022). DreamerV3 (Hafner et al., 2023) uses a fixed set of hyperparameters.

*Table 8.* **TD-MPC2 hyperparameters.** We use the same hyperparameters across all tasks. Certain hyperparameters are set automatically using heuristics.

| Hyperparameter | Value |
|---|---|
| **Planning** | |
| Horizon ($H$) | 3 |
| Iterations | 6 ($+2$ if $\|\mathcal{A}\| \geq 20$) |
| Population size | 512 |
| Policy prior samples | 24 |
| Number of elites | 64 |
| Minimum std. | 0.05 |
| Maximum std. | 2 |
| Temperature | 0.5 |
| Momentum | No |
| | |
| **Policy prior** | |
| Log std. min. | $-10$ |
| Log std. max. | 2 |
| | |
| **Replay buffer** | |
| Capacity | $1,000,000$ |
| Sampling | Uniform |
| | |
| **Architecture (5M)** | |
| Encoder dim | 256 |
| MLP dim | 512 |
| Latent state dim | 512 |
| Task embedding dim | 96 |
| Task embedding norm | 1 |
| Activation | LayerNorm + Mish |
| $Q$-function dropout rate | 1% |
| Number of $Q$-functions | 5 |
| Number of reward/value bins | 101 |
| SimNorm dim ($V$) | 8 |
| SimNorm temperature ($\tau$) | 1 |
| | |
| **Optimization** | |
| Update-to-data ratio | 1 |
| Batch size | 256 |
| Joint-embedding coef. | 20 |
| Reward prediction coef. | 0.1 |
| Value prediction coef. | 0.1 |
| Temporal coef. ($\lambda$) | 0.5 |
| $Q$-fn. momentum coef. | 0.99 |
| Policy prior entropy coef. | $1 \times 10^{-4}$ |
| Policy prior loss norm. | Moving $(5\%, 95\%)$ percentiles |
| Optimizer | Adam |
| Learning rate | $3 \times 10^{-4}$ |
| Encoder learning rate | $1 \times 10^{-4}$ |
| Gradient clip norm | 20 |
| Discount factor | Heuristic |
| Seed steps | Heuristic |

We set the discount factor $\gamma$ for a task using the heuristic

$$\gamma = \text{clip}(\frac{\frac{T}{5} - 1}{\frac{T}{5}}, \ [0.95, 0.995])$$ (7)

where $T$ is the expected length of an episode *after* applying action repeat, and clip constrains the discount factor to the interval $[0.95, 0.995]$. Using this heuristic, we obtain $\gamma = 0.99$ for DMControl ($T = 500$), which is the most widely used discount factor for this task domain. Tasks with shorter episodes are assigned a lower discount factor, whereas tasks with longer episodes are assigned a higher discount factor. All of the tasks that we consider are infinite-horizon MDPs with fixed episode lengths. We use individual discount factors (set using the above heuristic) for each task in our multitask experiments. For tasks with variable or unknown episode lengths, we suggest using an empirical mean length, a qualified guess, or simply $\gamma = 0.99$. While this heuristic is introduced in TD-MPC**2**, we apply the same discount factor for the TD-MPC and SAC baselines to ensure that comparison is fair across all task domains.

We set the seed steps $S$ (number of environment steps before any gradient updates) for a task using the heuristic

$$S = \max(5T, 1000)$$ (8)

where $T$ again is the expected episode length of the task after applying action repeat. We did not experiment with other heuristics nor constant values, but conjecture that Equation 8 will ensure that the replay buffer $\mathcal{B}$ has sufficient data for model learning regardless of episode lengths.

**Model configurations.** Our multitask experiments consider TD-MPC**2** agents with model sizes ranging from 1M parameters to 317M parameters. Table 9 lists the exact specifications for each of our model sizes. We scale the model size by varying dimensions of fully-connected layers, the latent state dimension $\mathbf{z}$, the number of encoder layers, and the number of $Q$-functions. We make no other modifications to the architecture nor hyperparameters across model sizes.

*Table 9.* **Model configurations.** We list the specifications for each model configuration (size) of our multitask experiments. *Encoder dim* is the dimensionality of fully connected layers in the encoder $h$, *MLP dim* is the dimensionality of layers in all other components, *Latent state dim* is the dimensionality of the latent representation $\mathbf{z}$, *# encoder layers* is the number of layers in the encoder $h$, *# Q-functions* is the number of learned $Q$-functions, and *Task embedding dim* is the dimensionality of $\mathbf{e}$ from Equation 2. TD-targets are always computed by randomly subsampling two $Q$-functions, regardless of the number of $Q$-functions in the ensemble. We did not experiment with other model configurations. **\***The default (`base`) configuration used in our single-task RL experiments has 5M parameters.

|                    | 1M  | 5M* | 19M  | 48M  | 317M |
|--------------------|-----|-----|------|------|------|
| Encoder dim        | 256 | 256 | 1024 | 1792 | 4096 |
| MLP dim            | 384 | 512 | 1024 | 1792 | 4096 |
| Latent state dim   | 128 | 512 | 768  | 768  | 1376 |
| # encoder layers   | 2   | 2   | 3    | 4    | 5    |
| # Q-functions      | 2   | 5   | 5    | 5    | 8    |
| Task embedding dim | 96  | 96  | 96   | 96   | 96   |

**Simplicial Normalization (SimNorm).** SimNorm is a simple method for normalizing the latent representation $\mathbf{z}$ by projecting it into $L$ fixed-dimensional simplices using a softmax operation (Lavoie et al., 2022). A key benefit of embedding $\mathbf{z}$ as simplices (as opposed to *e.g.* a discrete representation or squashing) is that it naturally biases the representation towards sparsity without enforcing hard constraints. Intuitively, SimNorm can be thought of as a *"soft"* variant of the vector-of-categoricals approach to representation learning proposed by Oord et al. (2017) (VQ-VAE). Whereas VQ-VAE represents latent codes using a set of discrete codes ($L$ vector partitions each consisting of a one-hot encoding), SimNorm partitions the latent state into $L$ vector partitions of continuous values that each sum to 1 due to the softmax operator. This relaxation of the latent representation is akin to softmax being a relaxation of the $\arg\max$ operator. While we do not adjust the temperature $\tau \in [0, \infty)$ of the softmax used in SimNorm in our experiments, it is useful to note that it provides a mechanism

for interpolating between two extremes. For example, $\tau \to \infty$ would force all probability mass onto single categories, resulting in the discrete codes (one-hot encodings) of VQ-VAE. The alternative of $\tau = 0$ would result in trivial codes (constant vectors; uniform probability mass) and prohibit propagation of information. SimNorm thus biases representations towards sparsity without enforcing discrete codes or other hard constraints. We implement the SimNorm normalization layer (Lavoie et al., 2022) using PyTorch-like notation as follows:

```python
def simnorm(self, z, V=8):
    shape = z.shape
    z = z.view(*shape[:-1], -1, V)
    z = softmax(z, dim=-1)
    return z.view(*shape)
```

Here, $z$ is the latent representation $\mathbf{z}$, and $V$ is the dimensionality of each simplex. The number of simplices $L$ can be inferred from $V$ and the dimensionality of $\mathbf{z}$. We apply a softmax (optionally modulated by a temperature $\tau$) to each of $L$ partitions of $\mathbf{z}$ to form simplices, and then reshape to the original shape of $\mathbf{z}$.

**TD-MPC baseline implementation.** We benchmark against the official implementation of TD-MPC available at https://github.com/nicklashansen/tdmpc. The default TD-MPC world model has approx. 1M trainable parameters, and uses per-task hyperparameters. We use the suggested hyperparameters where available (DMControl and Meta-World). For example, TD-MPC requires tuning of the number of planning iterations, latent state dimensionality, batch size, and learning rate in order to solve the challenging Dog and Humanoid tasks. Refer to their paper for a complete list of hyperparameters.

**DreamerV3 baseline implementation.** We benchmark against the official reimplementation of DreamerV3 available at https://github.com/danijar/dreamerv3. We follow the authors' suggested hyperparameters for proprioceptive control (DMControl) and use the S model size (20M parameters), as well as an update-to-data (UTD) ratio of 512. We use this model size and UTD for all tasks. Refer to their paper for a complete list of hyperparameters.

**SAC baseline implementation.** We follow the TD-MPC (Hansen et al., 2022) paper in their decision to benchmark against the SAC implementation from https://github.com/denisyarats/pytorch_sac, and we use the hyperparameters suggested by the authors (when available). For example, this includes tuning the latent dimension, learning rate, and batch size for the Dog and Humanoid tasks. Refer to their paper for a complete list of hyperparameters.

## I  EXTENDING TD-MPC2 TO DISCRETE ACTION SPACES

It is desirable to develop a single algorithm that excels at tasks with continuous and discrete action spaces alike. However, the community has yet to discover such an algorithm. While *e.g.* DreamerV3 (Hafner et al., 2023) has delivered strong results on challenging tasks with discrete action spaces (such as Atari and Minecraft), we find that TD-MPC2 produces significantly better results on difficult continuous control tasks. At the same time, extending TD-MPC2 to discrete action spaces remains an open problem. While we do not consider discrete action spaces in this work, we acknowledge the value of such an extension. At present, the main challenge in applying TD-MPC2 to discrete actions lies in the choice of planning algorithm. TD-MPC2 relies on the MPC framework for planning, which is designed for continuous action spaces. We believe that MPC could be replaced with a planning algorithm designed for discrete action spaces, such as MCTS (Coulom, 2007) as used in MuZero (Schrittwieser et al., 2020). It is also possible that there exists a way to apply MPC to discrete action spaces that is yet to be discovered (to the best of our knowledge), similar to how recent work (Hubert et al., 2021) has discovered ways to apply MCTS to continuous action spaces through sampling.

## J  TEST-TIME REGULARIZATION FOR OFFLINE RL

Our multi-task experiments revolve around training massively multi-task world models on fixed datasets that consist of a variety of behaviors, which is an offline RL problem. We do not consider any special treatment of the offline RL problem in the main paper, and simply train TD-MPC**2** agents without any additional regularization nor hyperparameter-tuning. However, we recognize that models may benefit from such regularization (conservative estimations) due to extrapolation errors when the dataset has limited state-action coverage and/or is highly skewed. Current offline RL algorithms are ill-suited for our problem setting, given that we aim to develop an algorithm that can seamlessly transition from massively multi-task offline pretraining to single-task online fine-tuning, without any changes in hyperparameters. Current offline RL techniques rely on *(1)* explicit or implicit conservatism in $Q$-value estimation which requires modifications to the training objective and empirically hampers online RL performance (Nakamoto et al., 2023), and *(2)* relies on a task-specific coefficient that balances value estimation and the regularizer. Instead, we propose to regularize the *planning procedure* of TD-MPC**2**, which can be done at test-time without any additional model updates. Concretely, we apply the test-time regularizer proposed by Feng et al. (2023), which penalizes trajectories with large uncertainty (as estimated by the variance in $Q$-value predictions) during planning. While this approach eliminates the need for training-time regularization, it still requires users to specify a coefficient that weighs estimated value relative to uncertainty for a trajectory, which is infeasible in a multi-task scenario where estimated values may differ drastically between tasks. To circumvent this issue, we propose a simple heuristic for automatically scaling the regularization strength at each timestep based on the (magnitude of) mean value predictions for a given latent state. Specifically, we estimate the uncertainty penalty at latent state $\mathbf{z}_t$ of a sampled trajectory as

$$u_t = c \cdot \mathrm{avg}([\hat{q}_1, \hat{q}_2, \ldots, \hat{q}_N]) \cdot \mathrm{std}([\hat{q}_1, \hat{q}_2, \ldots, \hat{q}_N]) \tag{9}$$

where $\hat{q}_n$ is a value prediction from $Q$-function $n$ in an ensemble of $N$ $Q$-functions, and $c$ is now a *task-agnostic* coefficient that balances return maximization and uncertainty minimization. The planning objective in Equation 6 is then redefined as

$$\mu^*, \sigma^* = \arg\max_{(\mu,\sigma)} \mathbb{E}_{(\mathbf{a}_t, \mathbf{a}_{t+1}, \ldots, \mathbf{a}_{t+H}) \sim \mathcal{N}(\mu, \sigma^2)} \tag{10}$$

$$\left[ \gamma^H Q(\mathbf{z}_{t+H}, \mathbf{a}_{t+H}) - u_{t+H} \right. \tag{11}$$

$$\left. + \sum_{h=t}^{H-1} \left( \gamma^h R(\mathbf{z}_h, \mathbf{a}_h) - u_h \right) \right], \tag{12}$$

using the definition of task-agnostic uncertainty in Equation 9. We conduct an experiment in which we apply our proposed test-time regularization to a 19M parameter TD-MPC**2** agent trained on the 80-task dataset, varying the regularization strength $c$. Results are shown in Table 10. Our results indicate that additional regularization (using our heuristic for automatic tuning) can indeed improve the average model performance for some values of $c$. Similar to what one would expect in a single-task setting, we find that large values of $c$ (strong regularization) decrease performance, whereas small (but $> 0$) values of $c$ tend to improve performance compared to TD-MPC**2** without regularization. Given that our heuristic with $c = 0.01$ leads to meaningful improvements across 80 tasks, we expect it to work reasonably well for other datasets as well, but leave this for future work.

*Table 10.* **Test-time regularization.** Normalized score of a 19M parameter TD-MPC**2** agent trained on the 80-task dataset, varying the regularization strength $c$ of our proposed test-time regularizer. We do not apply this regularization in any of our other experiments, and only include these results to inspire future research directions.

|                   | No reg. | $\mathbf{c = 0.001}$ | $\mathbf{c = 0.01}$ | $\mathbf{c = 0.1}$ |
| ----------------- | ------- | -------------------- | ------------------- | ------------------ |
| Normalized score  | 56.54   | 58.14                | **62.01**           | 44.13              |

## K   ADDITIONAL MULTI-TASK RESULTS

To provide further insights into the effect of data size and task diversity on TD-MPC**2** performance in a multi-task setting, we provide additional experiments on a 15-task subset of DMControl, selected at random. Results for TD-MPC**2** agents trained on 15 tasks, 30 tasks, and 80 tasks are shown in Figure 21. We observe that performance scales with model size across all three task suites, but numbers are higher across the board on the smallest dataset compared to similar capacity models trained on larger datasets. This makes intuitive sense, since model capacity remains the same while there are comparably fewer tasks to learn.

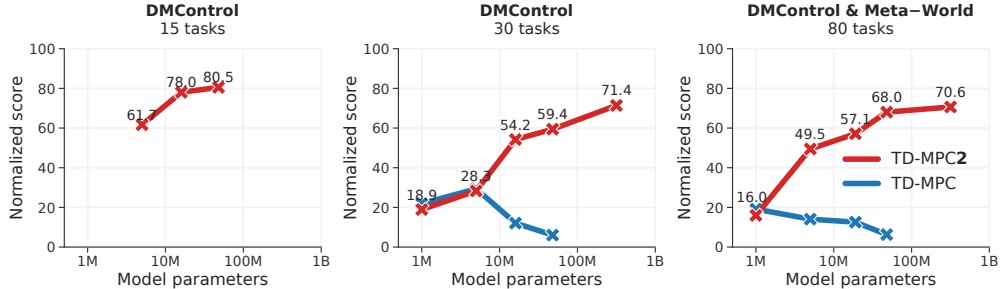

*Figure 21.* **Additional results on massively multi-task world models.** Normalized score as a function of model size on the 15-task, 30-task, and 80-task datasets. TD-MPC**2** capabilities scale with model size.

