# OpenReview forum: "TD-MPC2: Scalable, Robust World Models for Continuous Control"
_ICLR.cc/2024/Conference — ICLR 2024 spotlight_

### Official Review · Reviewer_jMX1 · 2023-10-18

**Soundness:** 3 good
**Presentation:** 4 excellent
**Contribution:** 4 excellent
**Rating:** 8
**Confidence:** 5

**Summary:**

This paper proposes TDMPC2, which is a series of improvements over TDMPC for model-based RL in continuous domains. As far as I can tell, the following are the key differences:

** Changes to make the algorithm multi-task **
- Adding e (learned task embedding) to all components of the TOLD model.
- Zero-padding invalid dimensions for observations and actions.
- Handling differing reward magnitude by using cross-entropy instead of squared error for reward and value prediction losses.

** Architectural changes **
- SimNorm
- Using an ensemble of Qs

** Other changes **
- Including an entropy bonus in policy optimization objective, following maximum entropy literature.

As far as I can tell, the planning algorithm remains unchanged.

(If I have missed anything important, please let me know.)

The proposed modifications lead to large improvements over TDMPC when scaled up to larger model sizes, and enable multi-task learning.

**Strengths:**

- The introductory figure on page 1 is a great idea, it works very well at describing the results at-a-glance and gets the reader interested to see more.
- The paper is organized well and provides clear descriptions of the approach.
- Experiments are thorough and convincing. Results are extremely favorable compared to other SOTA approaches like DreamerV3 and TDMPC. Ablations are also provided to delineate the impact of each modification.
- Figure 7 (right) is very interesting, thank you for providing this visualization of the task embeddings.
- Code, data, models are provided open-source. Training GPU costs being provided in Table 1 is also a plus.

**Weaknesses:**

I believe this paper should be accepted on the merit of the strong results and presentation. However, there are some major concerns I have that would be great to address in the next revision.

- My main suggestion, *highly recommended*, is for the authors to include in the main text a list of the differences from TDMPC. There will be many readers (like myself) who are familiar with TDMPC, and we'd like to be able to understand the differences at a glance. The way the paper is currently written, the authors have described the full algorithm from first principles, but this mixes together their contributions versus TDMPC's contributions. At best, this causes confusion, while at worst, a reader may incorrectly attribute credit for an idea to this paper rather than the original TDMPC paper. A concrete example is the section on the planning algorithm; rather than a  vague sentence at the end saying "Refer to [the TDMPC paper] for more details about the planning procedure" (which hides whether or not there are any differences), it would be good to start off with a direct statement like "The planning procedure is unchanged from [the TDMPC paper]; we re-describe it here so that this paper can be self-contained." Or, if there are minor changes, you can describe them candidly.

- Another smaller suggestion: There are some parts of the paper that feel like too much overselling of what is ultimately a couple of modifications from TDMPC. The sentence which got me feeling this way was "TDMPC2 marks the beginning of a new era for model-based RL". It's great if the authors feel that way about their own work, but in a scientific paper I would prefer the language to remain neutral, and to allow the reader to form their own judgment about these matters.

- I have included several smaller questions below that should be addressed in the next revision as well.

**Questions:**

- How does the DreamerV3 baseline work, doesn't that algorithm expect discrete action spaces (as discussed in Section 5)?
- I'd like some clarifications on the task embeddings e. Are they just trainable vectors that get passed as input to all components of the TOLD model? What forces them to be meaningfully diverse from each other in a way that is consistent with the semantics of the tasks? Is there some extra term in the loss function for them?
- "minimum of two randomly sub-sampled Q-functions" Why not just take the minimum over all 5?
- "reward and value prediction as a discrete regression" Are there limitations arising from this due to the resulting lack of expressivity? Or is the discretization fine enough that a wide enough range of value functions can be effectively learned? What are the tradeoffs of increasing or decreasing the granularity of this discretization?
- The SimNorm contribution is interesting. Is it fair to think of it as a different method of achieving the same effect as the vector-of-categoricals approach in Appendix G of the Director paper (https://arxiv.org/abs/2206.04114)?
- "extending TD-MPC2 to discrete action spaces remains an open problem" What are the challenges here?

---

> ### Author Response · Authors · 2023-11-21
> **Author response to reviewer jMX1**
>
> We thank the reviewer for their valuable feedback. We address your comments in the following.
>
> ----
>
> **Q:** My main suggestion, *highly recommended*, is for the authors to include in the main text a list of the differences from TDMPC.
>
> **A:** We fully agree. We actually provide a full list (~1 page) of differences between TD-MPC and TD-MPC2 in Appendix A, but regrettably did not reference it in the method section. We have added a reference to this list in the updated manuscript.
>
> ----
>
> **Q:** Another smaller suggestion: There are some parts of the paper that feel like too much overselling of what is ultimately a couple of modifications from TDMPC.
>
> **A:** We agree with the sentiment of this, and have updated the manuscript to describe the algorithm and results in a more neutral tone. Thank you for the suggestion!
>
> ----
>
> **Q:** How does the DreamerV3 baseline work, doesn't that algorithm expect discrete action spaces (as discussed in Section 5)?
>
> **A:** DreamerV3 (as well as previous iterations) can be applied to either discrete or continuous action spaces by changing the parameterization of its latent policy (Gaussian vs. categorical), similar to SAC. However, DreamerV3 had previously only been benchmarked on an easy subset of the DMControl tasks, with all other tasks being discrete. We find that DreamerV3 performs quite well on this set of tasks, but performs significantly worse when applied to other continuous control tasks – both harder DMControl tasks, as well as new domains.
>
> ----
>
> **Q:** I'd like some clarifications on the task embeddings e. Are they just trainable vectors that get passed as input to all components of the TOLD model? What forces them to be meaningfully diverse from each other in a way that is consistent with the semantics of the tasks? Is there some extra term in the loss function for them?
>
> **A:** Correct – the task embeddings are simply learnable vectors. We maintain a dictionary of vectors (one for each task), and condition all components on the retrieved vector(s). They are trained end-to-end together with other components using the objective from Equation 3. Intuitively, the semantic meaning of task embeddings likely “emerges” because it is the most straightforward way for the model to distinguish different tasks and embodiments. Our only constraint (which we do find to be moderately important) is that we constrain the task embeddings to have a norm of 1. We ablate this in Appendix F, both in terms of normalized score and semantic coherence (T-SNE visualizations).
>
> ----
>
> **Q:** "minimum of two randomly sub-sampled Q-functions" Why not just take the minimum over all 5?
>
> **A:** While this design choice may seem somewhat arbitrary at first glance, there is some evidence that taking the minimum/average over a large ensemble of Q-functions can actually be detrimental to data-efficiency as it propagates biases from the target Q-function. For example, REDQ by Chen et al. [1] demonstrates this phenomenon for SAC trained on a set of continuous control tasks. Additionally, taking the minimum over the full ensemble would also be slower than when sub-sampling.
>
> [1] Chen et al., Randomized Ensembled Double Q-Learning (https://arxiv.org/abs/2101.05982)
>
> ---
>
> **Q:** The SimNorm contribution is interesting. Is it fair to think of it as a different method of achieving the same effect as the vector-of-categoricals approach in Appendix G of the Director paper (https://arxiv.org/abs/2206.04114)?
>
> **A:** To some extent. The high-level motivation is similar: constraining the latent state in some way is beneficial for learning. DreamerV2/V3/Director factorizes the representation as multiple categoricals that function as discrete codes (originally proposed in VQ-VAE), whereas SimNorm can be viewed as a “soft” variant in the sense that its representation is factorized as multiple simplices (each sub-vector sums to 1). Intuitively, SimNorm *biases* the representation towards sparsity, whereas vector-of-categorical forces the representation to be sparse, in the same way that Softmax can be interpreted as a relaxation of argmax. While we do not adjust the temperature of the SimNorm Softmax in our experiments, it does provide a mechanism for interpolating between the same two extremes: infinite temperature would result in a discrete (one-hot) code, whereas zero temperature would result in constant (uniform) vectors. We have added an extended discussion on this in Appendix H of the updated manuscript.
>
> ----
>
> Please do not hesitate to let us know if you have any additional comments.

---

> > ### Comment · Reviewer_jMX1 · 2023-11-22
> > **response to rebuttal**
> >
> > Thank you, I hadn't seen App A but the new reference to it will surely help.
> >
> > I also missed this part in the main paper, though looking back, now I do see the relevant sentence in Sec 3.3: "Our only constraint (which we do find to be moderately important) is that we constrain the task embeddings to have a norm of 1." It might be good to highlight that sentence a bit more somehow.
> >
> > I guess you may have missed my last question: "extending TD-MPC2 to discrete action spaces remains an open problem" What are the challenges here?"
> >
> > In any case, this is a good rebuttal and I feel good about this paper. Given my score of 8 I am inclined to leave it as is and will advocate for acceptance.

---

> > > ### Author Response · Authors · 2023-11-22
> > > **Author response to reviewer jMX1**
> > >
> > > Thank you for acknowledging our response, for the kind words, and for pointing out that we missed your last question! Apologies for that.
> > >
> > > To answer your question: we recognize that bridging continuous and discrete action spaces is an important problem that warrants further research. The challenge mainly lies in the choice of planning algorithm. TD-MPC2 relies on the Model Predictive Control (MPC) framework for planning, which is designed for continuous action spaces. We believe that MPC could be replaced with a planning algorithm designed for discrete action spaces, such as MCTS (used in MuZero), although that is not the focus of this paper. It is also possible that there exists a way to apply MPC to discrete action spaces that is yet to be discovered (to the best of our knowledge), similar to how researchers have discovered ways to apply MCTS to continuous action spaces through adaptive discretization and clustering methods. We have added an extended discussion of this topic to Appendix I of our updated manuscript, which should be useful for readers who want to extend our algorithm to discrete action spaces.
> > >
> > > Regarding the task embedding norm: we agree that the paper is quite dense and that it is easy to miss some details on first read due to the sheer number of design choices and experiments. We hope that providing extensive details in the appendices + open-sourcing code can help in this regard. Either way: in response to your comment, we have updated the relevant paragraph in Sec. 3.3 to provide slightly more context, thus making it harder to miss.
> > >
> > > Thanks again for bringing it up!

---

### Official Review · Reviewer_8R3o · 2023-10-22

**Soundness:** 3 good
**Presentation:** 4 excellent
**Contribution:** 3 good
**Rating:** 8
**Confidence:** 5

**Summary:**

This paper introduces TD-MPC2, which incorporates improvements over a strong model-based RL baseline called TD-MPC. The improvements focus on generalization across tasks and action spaces. Experiment results show that TD-MPC2 is a viable method for continuous control tasks and is less sensitive to hyperparameters, and that it benefits from scaling both model and data sizes.

**Strengths:**

1. TD-MPC2 is a strong model-based RL method that doesn't require significant task-specific hyperparameters tuning.
2. The paper is very well-written.
3. Extensive experiment results.

**Weaknesses:**

The study on multi-task learning is a bit unsatisfying, mainly due to the limited (and somewhat artificial) setup of the training data which was obtained from the replay buffers of the single-task TD-MPC2 agents. It would be great if the authors could consider extend their studies to more realistic training data distribution, e.g. mixture of few expert demonstrations and massive suboptimal data as prevalent in robotics.

**Questions:**

On multi-task setting:
1. Have you considered training the multi-task model online?
2. Have you explored other dataset setups, i.e. different random to expert ratios?
3. How about the impacts of task diversity, i.e. ablation on number of tasks in the training data?
4. How do the multitask model perform compared to single-task models on each individual tasks?
5. Have you considered leveraging offline RL techniques when learning from the fixed multi-task dataset? Do you think they can help?

Other questions:
1. The observation that TD-MPC's performance decreases as model and data sizes increase is very interesting. Do you have any hypothesis of why that's the case?
2. How did you do hyperparameter search?
3. Does TD-MPC2 benefit from model size scaling on single-task setting?

---

> ### Author Response · Authors · 2023-11-21
> **Author response to reviewer 8R3o (1/3)**
>
> We thank the reviewer for their valuable feedback. We address your comments in the following.
>
> ----
>
> **Q:** Have you considered training the multi-task model online?
>
> **A:** Yes, we considered that, but ultimately decided against it to (1) avoid the significant wall-time overhead of simulation, and (2) to eliminate additional factors of variation such as exploration that might influence our scaling results. However, given the strong single-task online RL results, we do not expect conclusions to change drastically based on this experimental design choice.
>
> ----
>
> **Q:** Have you explored other dataset setups, i.e. different random to expert ratios? Have you considered leveraging offline RL techniques when learning from the fixed multi-task dataset? Do you think they can help?
>
> **A:** We did not experiment with other data distributions, since we wanted to focus on building an RL algorithm that works *at scale*. Offline RL problem settings and techniques are somewhat orthogonal to realizing this goal, since offline RL techniques focus specifically on mitigating extrapolation errors in small or highly skewed datasets. For example, prior work [1] shows that standard online RL algorithms without explicit/implicit conservatism (*i.e.*, without any of the popular offline RL techniques) outperform offline RL methods when data is abundant and covers a wider state-action distribution than what the replay buffer of an agent would typically provide. Thus, we would expect vanilla TD-MPC2 to outperform TD-MPC2 with any of the common offline RL techniques added when evaluating on *maximally diverse* data. Our current setting uses replay data, and therefore falls somewhere in between hard offline RL (narrow data distribution, mainly suboptimal data) and ideal data (uniform state-action distribution coverage). Whether TD-MPC2 (as well as any other RL algorithm) benefits or suffers from added conservatism thus depends entirely on where the specific dataset falls on this spectrum.
>
> However, to address your question experimentally, we conduct an additional experiment where we enforce conservatism in a 19M parameter TD-MPC2 agent trained on our 80-task dataset. It is worth noting that most existing offline RL techniques (1) regularize predictions at *training-time*, and (2) require task-specific tuning of the coefficient that balances return maximization and conservatism / uncertainty, which combined makes it difficult to apply in a multitask setting like ours. Instead, we choose to regularize the *planning* of TD-MPC2 at *test-time* (*i.e.*, we apply it to our existing checkpoint without any additional model updates) using the technique proposed by Feng et al. [2], which discourages trajectories with large uncertainty in the return estimation. Like most other offline RL methods, however, this method was proposed for a single-task setting and requires per-task tuning of the regularizer. To circumvent this issue, we propose a simple heuristic for automatically scaling the regularization strength at each timestep based on the (magnitude of) mean value predictions for a given latent state. We provide a more precise mathematical definition in Appendix J of the updated manuscript. Our results indicate that additional regularization (and using our heuristic for automatic tuning) can indeed improve the average model performance for some values of our task-agnostic coefficient $c$, but we view this finding as orthogonal to the main contributions of our paper. Similar to what one would expect in a single-task setting, we find that large values of c (strong regularization) decrease performance, whereas small (but $>0$) values of c tend to improve performance compared to TD-MPC2 with no regularization. Given that our heuristic with $c = 0.01$ leads to meaningful improvements across 80 tasks, we expect it to also work reasonably well for other datasets. We have added this experiment + discussion to Appendix J, and thank the reviewer for the suggestion.
>
> | 80 tasks, 19M params | **No reg.** | **c = 0.001** | **c = 0.01** | **c = 0.1** |
> |----------------------|-------------|---------------|--------------|-------------|
> | Normalized score     | 56.54       | 58.14         | **62.01**        | 44.13       |
>
>
> [1] Yarats et al., Don’t Change the Algorithm, Change the Data: Exploratory Data for Offline Reinforcement Learning, arXiv 2022 (https://arxiv.org/abs/2201.13425)
>
> [2] Feng et al., Finetuning Offline World Models in the Real World, CoRL 2023 (https://arxiv.org/abs/2310.16029)

---

> > ### Author Response · Authors · 2023-11-21
> > **Author response to reviewer 8R3o (2/3)**
> >
> > **Q:** How about the impacts of task diversity, i.e. ablation on number of tasks in the training data?
> >
> > **A:** This is a very interesting question that unfortunately is difficult to give a conclusive answer to with the current task suites available to us. I suspect that the conclusion will be different for our current 80 tasks vs. e.g. a hypothetical future task suite consisting of 8,000 diverse tasks. However, we presently do observe similar “scaling laws” for both our 30-task dataset based on DMControl, and our 80-task dataset based on DMControl + Meta-World, when we control for model size. To give you a slightly more conclusive answer, we train another set of multi-task models (5M, 19M, 48M parameters) on a 15-task subset of our DMControl task suite. Our results are shown [here](https://i.imgur.com/dToYApX.png). We observe that performance still scales with model size on our smaller task suite, but that normalized scores are higher across the board. Notably, the 5M parameter model fares far better on the smaller 15-task dataset than with 30 tasks. This makes intuitive sense, since model capacity remains the same while there is comparably less knowledge to learn. However, we remark that (1) it is difficult to directly compare model performance across task suites since we cannot fully control for task difficulty and “diversity”, and (2) the 48M parameter model trained on 15 tasks has not yet fully converged; we expect that it will continue to improve a bit with more training. The 5M and 19M models are trained to convergence. We will be happy to provide an update once the model has converged.
> >
> > ----
> >
> > **Q:** How do the multitask model perform compared to single-task models on each individual tasks?
> >
> > **A:** To answer this question, we compute a single normalized score as the average score of final single-task checkpoints trained on each of 80 tasks included in our largest multitask dataset. Note that these single-task models are trained **online**, and are thus not limited by the offline RL setting (fixed dataset) that our multitask model is trained with. Additionally, single-task models are trained to convergence and thus represent a (soft) upper bound on the multitask performance. This procedure gives us a score of 91.8 compared to 70.6 for our largest multitask model, without any offline RL techniques. We conjecture that this gap is due to (1) limited similarity between tasks, (2) limited data diversity for offline RL, and (3) insufficient model parameters to accurately model all 80 tasks. However, most works on multi-task model-based RL consider significantly fewer tasks, so we believe that building an algorithm that scales *at all* will be incredibly useful to the community; both as is, and as a basis for future research on large-scale RL.
> >
> > ----
> >
> > **Q:** The observation that TD-MPC's performance decreases as model and data sizes increase is very interesting. Do you have any hypothesis of why that's the case?
> >
> > **A:** Most contemporary RL algorithms rely on small models or policies (~1M parameters) and do not scale reliably with more model capacity or data. While the research community does not have a conclusive answer for why this is, we believe that the main culprit is non-stationarity and poor signal-to-noise ratio in the learning objective (Q-learning in particular), which are generally absent in supervised learning. As for the specifics of TD-MPC, we do observe that both layer weights and latent states $z$ trend towards large values when not constrained in any way, resulting in very large gradient norms (see Appendix G in our manuscript) and training instabilities. TD-MPC2 mitigates this problem through a series of careful design choices, virtually eliminating training instabilities. While some design choices are unique to the TD-MPC algorithm and MBRL literature (e.g. SimNorm, planning), we do believe that some of our design choices (e.g. LayerNorm, discrete regression) could also improve stability of other RL algorithms such as SAC in the future.
> >
> > ----
> >
> > **Q:** How did you do hyperparameter search?
> >
> > **A:** We did not do any noteworthy hyperparameter search – most hyperparameters are adopted from TD-MPC without modification. Instead, we set out to build an algorithm that is robust enough to task variations that we do not need to tune hyperparameters in the first place. This is achieved through careful design of the architecture and model objective. We expect this approach to be a more viable solution in the long term, especially as other researchers and practitioners in the community start to apply TD-MPC2 to new task domains that were not considered in our paper.
> >
> > ----
> >
> > (2/3)

---

> > > ### Author Response · Authors · 2023-11-21
> > > **Author response to reviewer 8R3o (3/3)**
> > >
> > > **Q:** Does TD-MPC2 benefit from model size scaling on single-task setting?
> > >
> > > **A:** We have run additional ablations on the choice of model size in response to your question; our results are shown [here](https://i.imgur.com/t94RKfO.png). Ablations are conducted on the same three tasks as in previous ablations: Dog Run, Humanoid Walk (DMControl), and Pick YCB (ManiSkill2). We observe that different model sizes yield similar data-efficiency and asymptotic performance across model sizes in a single-task setting, with the 5M parameter model being marginally better on average. We attribute this to the fact that (1) small models are generally sufficient for modeling a single task, and (2) larger models may require more gradient updates to fit the data which we do not control for in this experiment. More research is needed to determine whether there is tangible benefit in training large models on small datasets. However, it is notable that the algorithmic robustness of TD-MPC2 does prevent the larger models from diverging during training, which is otherwise commonly observed in TD-MPC and other contemporary RL algorithms (as evidenced e.g. in our multi-task experiments).
> > >
> > > ----
> > >
> > > Please do not hesitate to let us know if you have any additional comments.

---

> > > > ### Author Response · Authors · 2023-11-22
> > > > **Following up**
> > > >
> > > > Hi again,
> > > >
> > > > The discussion phase is coming to an end (today EOD), and we thus kindly ask you to review our response to your comments. Please do not hesitate to let us know if you have any follow-up questions! If not, we would be grateful if you would consider updating your score to reflect that the issues have been addressed.
> > > >
> > > > Best,
> > > >
> > > > Authors of TD-MPC2 (submission 559)

---

> > > > > ### Comment · Reviewer_8R3o · 2023-11-22
> > > > >
> > > > > Thank you for the detailed response. My main concerns and questions have been mostly addressed. I've thus increased my score to accept. Good work!

---

### Official Review · Reviewer_cggF · 2023-11-01

**Soundness:** 3 good
**Presentation:** 3 good
**Contribution:** 3 good
**Rating:** 8
**Confidence:** 3

**Summary:**

The paper presents TD-MPC2, which is an upgrade for the model-based RL algorithm TD-MPC. The method is evaluated across a large number of RL tasks including multi-task RL benchmarks. The results show a significant boost of performance compared to other model-based RL baselines especially for Multi-task RL.

**Strengths:**

1. The method is evaluated on a large number of tasks in different domains. It's nice to see the huge boost of performance in multi-task scenarios.

2.  The model checkpoints and code are provided. All the hyperparameters are also provided in the appendix.

3. TD-MPC is one of the state-of-the-art model-based RL algorithms so it's great to see an upgrade to it.

**Weaknesses:**

From the results in appendix, it seems that in single task cases, TD-MPC2 is better than the original TD-MPC in terms of only a few more complicated tasks, is that true?

The novelty is somewhat limited as the modifications to TD-MPC seem to be all from some other existing methods.

**Questions:**

See Weaknesses.

---

> ### Author Response · Authors · 2023-11-21
> **Author response to reviewer cggF**
>
> We thank the reviewer for their valuable feedback. We address your comments in the following.
>
> ----
>
> **Q:** From the results in appendix, it seems that in single task cases, TD-MPC2 is better than the original TD-MPC in terms of only a few more complicated tasks, is that true?
>
> **A:** Correct; TD-MPC performs well on many of the tasks that we consider. TD-MPC2 performs comparably on most tasks, and improves significantly over TD-MPC on difficult tasks (Dog, Humanoid, Pick YCB, Pick Place Wall, MyoSuite, etc.). However, TD-MPC2 does so **without** task-specific hyperparameters, whereas TD-MPC requires hyperparameter tuning for each embodiment / domain. We believe that developing an algorithm that can learn diverse tasks without any tuning is extremely valuable. We would also like to reiterate that the success of our multi-task scaling experiments (which is unprecedented in literature) are a direct result of the design choices that also lead to our tuning-free single-task results.
>
> ----
>
> **Q:** The novelty is somewhat limited as the modifications to TD-MPC seem to be all from some other existing methods.
>
> **A:** Respectfully, this can be said about many of the RL papers that have had a profound impact on the field; Rainbow, EfficientZero, RAD, CURL, DrQ, Dreamer all come to mind, as well as the DroQ algorithm brought up by reviewer [BqZp](https://openreview.net/forum?id=Oxh5CstDJU&noteId=xCoQaNp0RE). We believe that there is value in pushing the boundaries of what can be done with existing algorithms, through a series of carefully made design choices informed by large-scale experimental results backed by existing literature. Additionally, we would like to reiterate that TD-MPC2 is the **first** RL algorithm to (1) learn continuous control tasks from diverse domains without any tuning, and (2) scale to large models trained on massively multi-task datasets, **neither** of which are trivial to achieve.
>
> ----
>
> Please do not hesitate to let us know if you have any additional comments.

---

> > ### Comment · Reviewer_cggF · 2023-11-21
> >
> > Thanks for your reply. I understand and agree with the authors' perspectives in both comments. Upon further reading, I have one more question (minor): I found the proposed way of transforming the reward prediction problem into a discrete regression for multitask learning very interesting. Have the authors perform any ablation study related to this approach? Like comparing to just use the sparse reward or manually normalize the reward for different tasks to [-1,1].

---

> ### Author Response · Authors · 2023-11-21
> **Author response to reviewer cggF**
>
> Thank you for acknowledging our response! We are glad to hear that our response has addressed your primary concerns.
>
> **Regarding alternatives to discrete regression:**
>
> Firstly, we'd like to clarify that while the problem of inconsistent reward ranges between tasks may be particularly relevant for multi-task learning, it is also problematic in single-task learning when one wants to use fixed hyper-parameters, since magnitude of reward/value directly affects the magnitude of gradients and individual loss terms. You will find that many open-source implementations of popular algorithms scale rewards manually, and that the scale depends both on the specific task and algorithm used. For example, the [minimalRL](https://github.com/seungeunrho/minimalRL/tree/master) (2.6k stars) implementations of [DQN](https://github.com/seungeunrho/minimalRL/blob/c8efed8481e3cd40e9739cfde220a55522555b57/dqn.py#L97) and [SAC](https://github.com/seungeunrho/minimalRL/blob/c8efed8481e3cd40e9739cfde220a55522555b57/sac.py#L151) scale rewards by $0.01$ and $0.1$, respectively, for the same task (Cartpole). While we cannot possibly know how the code authors decided on these values, it is likely that they were discovered simply by exhaustively trying different values. It is thus very desirable to build an algorithm that eliminates the need for such tuning altogether.
>
> Either way, discrete regression is one of the most straightforward ways to address this problem. There are other options, but they each have their own drawbacks:
>
> - **Sparse rewards:** Binarizing the reward signal based on either success criteria or reward thresholding addresses the initial problem of inconsistent reward ranges, but would drastically reduce the available learning signal for tasks in which rewards are highly shaped (common in locomotion), and also makes the exploration problem significantly harder (no learning signal unless a success state / reward threshold is reached). Lastly, there exists many tasks for which defining a binary success flag is difficult; while it may work for e.g. pick-and-place style manipulation tasks where success is tied to concrete states (is the object at the target location?), this may not work for other tasks such as learning locomotion behaviors (abstract goals extended in time, such as "run indefinitely" or "make a backflip") for which defining a reward threshold would require knowledge about (1) the exact definition of the reward function and its range of possible values for each individual task, and (2) which reward threshold constitutes "successful" behavior, neither of which are feasible to define at scale.
> - **Manual normalization:** Normalizing rewards manually is certainly an option, but requires privileged information about the reward function in question and its possible values, and is not feasible for a human to do beyond maybe a dozen tasks. We kind of addressed this approach in our first paragraph, so we'll keep this one brief.
> - **Automatic normalization:** Rather than specifying reward scaling coefficients manually, it is also possible to automatically scale rewards based on running statistics. However, this approach has seen limited use in existing literature since running statistics introduce non-stationarity in the prediction problem, and is sensitive to outliers (infrequent very small/large rewards).
>
> One last point that also favors our discrete regression approach is that -- even with normalized single-step rewards -- *value* targets may still be on different scales for different tasks. This is true for our considered task domains, since they differ both in episode length, distribution of rewards within an episode (rewards are not evenly distributed within successful episodes), discount factor (prediction horizon), and sparsity (some of our tasks use binary rewards whereas others use shaped rewards).
>
> We hope that our responses have addressed the reviewer's concerns. Please let us know if not! Otherwise, we kindly ask the reviewer to consider raising their score accordingly.

---

> > ### Comment · Reviewer_cggF · 2023-11-22
> >
> > Thanks for the detailed response. I have increased my score to a clear accept.

---

### Official Review · Reviewer_BqZp · 2023-11-01

**Soundness:** 3 good
**Presentation:** 4 excellent
**Contribution:** 4 excellent
**Rating:** 8
**Confidence:** 4

**Summary:**

This paper introduces TD-MPC2, an extension of the TD-MPC algorithm that introduces a set of improvements. TD-MPC is a model-based algorithm that performs Model-Predictive Control (MPC) in the latent space of a learned world model, and bootstraps the value of the final state on the planning horizon with a learned value function obtained via a model-free RL algorithm.
The introduced improvements include novel architectural designs (e.g., Layer Norm), using maximum-entropy RL as the policy prior, and, most importantly, a multi-task world model capable of learning the dynamics of several tasks with a single set of parameters. The experiments evaluate TD-MPC2 with TD-MPC, as well as with model-based and model-free SOTA algorithms, on several multi-task settings.

**Strengths:**

* The paper is well-written and organized, and includes a thorough discussion of the relevant related works.

* The experimental results showcase impressive performance gains over the state-of-the-art on a wide range of robotics tasks (e.g., DM-Control, Meta-World) and settings (e.g., multi-task, few-shot learning).

* The authors provide model checkpoints, datasets, and code for training and evaluating their proposed method. This significantly facilitates reproducing the experimental results and extending the introduced method.

**Weaknesses:**

* The method is not directly applicable to domains with discrete action spaces.

* Because the proposed method relies on MPC, it inherits a few of its drawbacks (e.g., decision-time computational overhead, difficulty handling multi-modal transitions).

* A few algorithmic decisions of the method could be better motivated, e.g., by giving an intuitive explanation of why they are expected to bring benefits. For instance, the discussion on why using SimNorm and how it biases the representation towards sparsity could be improved with an intuitive explanation of Eq. (5).

**Questions:**

Below, I have a few question and constructive feedback to the authors:

1) Why did you decide to use the Mish activation function? Did it provide significant improvements compared to other commonly used activations, e.g., ReLU?

2) I am not sure that it is possible to claim that MPC is a closed-loop policy, even when it considers the terminal value function at time step $H$. For instance, if Eq. (6) was maximized over a sequence of policies $(\pi_t,...,\pi_{t+h})$ instead of a sequence of actions $(a_{t},...,a_{t+h})$, it could result in a higher value. See Eq. (2) of [1] as an example. I believe this could lead to sub-optimal behavior in highly stochastic or multi-modal environments.

3) Recently, [2] also showed performance gains via Layer Norm and Dropout. Is this related to how Layer Norm and Dropout were used on TD-MPC2? I suggest discussing these techniques in more depth.

4) “SAC and DreamerV3 are prone to numerical instabilities in Dog tasks”. Could you elaborate on which numerical instabilities, and why TD-MPC2 can avoid them?

5) “At the same time, extending TD-MPC2 to discrete action spaces remains an open problem.”
I suggest elaborating what are the challenges involved in applying TD-MPC2 on discrete action spaces.

Minor:
- Below Eq. (3), it is missing a $\gamma$ in the TD-target $q_t = r_t + \gamma Q$.

[1] Online Planning with Lookahead Policies. Yonathan Efroni et al. NeurIPS 2020.

[2] Dropout Q-Functions for Doubly Efficient Reinforcement Learning. Hiraoka et al. ICLR 2022.

---

> ### Author Response · Authors · 2023-11-21
> **Author response to reviewer BqZp (1/2)**
>
> We thank the reviewer for their valuable feedback. We address your comments in the following.
>
> ----
>
> **Q:** Why did you decide to use the Mish activation function?
>
> **A:** We observe that Mish activations lead to slightly more consistent results compared to ReLU (common in RL) and ELU (used in TD-MPC) activations. However, TD-MPC2 works well with either of them. We have run additional ablations on the choice of activation function in response to your question; links to results are shown [here](https://i.imgur.com/BG5novj.png). Ablations are conducted on the same three tasks as in previous ablations: Dog Run, Humanoid Walk (DMControl), and Pick YCB (ManiSkill2). Results indicate that performance (return / success rate) is similar across activations, but that Mish leads to smaller and more stable gradients compared to the alternatives. Computational overhead of Mish vs. ReLU/ELU is negligible, so we decided to use Mish activations. We have added these additional ablations to our updated manuscript.
>
> ----
>
> **Q:** Recently, [2] also showed performance gains via Layer Norm and Dropout. Is this related to how Layer Norm and Dropout were used on TD-MPC2?
>
> **A:** Short answer: yes. While TD-MPC2 does not use a Transformer architecture per se, we indeed find that TD-MPC2 benefits from many of the same design choices that have been instrumental for scaling Transformers to large datasets. Examples of such design choices include MLPs with LayerNorm, Dropout, Softmax (key to our proposed SimNorm representation), and multiple prediction heads. The referenced paper on Dropout Q-functions [2] shows that Q-learning can benefit from one of such components (Dropout), whereas we demonstrate that all of these components complement each other when carefully applied to the TD-MPC algorithm (along with our other proposed changes).
>
> ----
>
> **Q:** “SAC and DreamerV3 are prone to numerical instabilities in Dog tasks”. Could you elaborate on which numerical instabilities, and why TD-MPC2 can avoid them?
>
> **A:** While it is difficult to give a conclusive answer to why SAC and DreamerV3 experience numerical instabilities for this set of tasks, we can attempt to extrapolate our knowledge about the tasks and instabilities in TD-MPC to these two algorithms. Firstly, it is worth noting that the Dog tasks have a very large action space compared to other locomotion tasks (38 continuous dimensions vs. e.g. 21 for Humanoid and 6 for Walker). We observe that SAC and DreamerV3 also fail to learn in MyoSuite, which has a similarly large action space (39 dimensions). Based on our ablations in Figure 9 (actor) + the fairly strong performance of TD-MPC1 on these tasks, we believe that planning plays a key role in tasks with large action spaces. Secondly, we observe that rewards increase for SAC during its initial stages of training, but then diverge without recovery. This is qualitatively similar to what we observe in TD-MPC, for which gradients grow extremely large before ultimately diverging. TD-MPC2 improves training stability through a series of changes to the architecture and objective, which result in consistently small gradients through training.
>
> ----
>
> **Q:** I suggest elaborating what are the challenges involved in applying TD-MPC2 on discrete action spaces.
>
> **A:** Good suggestion. We recognize that bridging continuous and discrete action spaces is an important problem. The challenge mainly lies in the choice of planning algorithm. TD-MPC2 relies on the Model Predictive Control (MPC) framework for planning, which is designed for continuous action spaces. We believe that MPC could be replaced with a planning algorithm designed for discrete action spaces, such as MCTS (used in MuZero), although that is not the focus of this paper. It is also possible that there exists a way to apply MPC to discrete action spaces that is yet to be discovered (to the best of our knowledge), similar to how researchers have discovered ways to apply MCTS to continuous action spaces through adaptive discretization and clustering methods. We have added this discussion to Appendix I of our updated manuscript.
>
> ----
>
> (1/2)

---

> > ### Author Response · Authors · 2023-11-21
> > **Author response to reviewer BqZp (2/2)**
> >
> > **Q:** A few algorithmic decisions of the method could be better motivated, e.g., by giving an intuitive explanation of why they are expected to bring benefits. For instance, the discussion on why using SimNorm and how it biases the representation towards sparsity could be improved with an intuitive explanation of Eq. (5).
> >
> > **A:** Agreed! We have added a new subsection on the intuition behind SimNorm in Appendix H of the updated manuscript. Intuitively, SimNorm can be thought of as a "soft" variant of the vector-of-categoricals approach to representation learning proposed in VQ-VAE. Whereas VQ-VAE represents latent codes using a set of discrete codes ($L$ vector partitions each consisting of a one-hot encoding), SimNorm partitions the latent state into $L$ vector partitions of continuous values that each sum to $1$ due to the softmax operator. This relaxation of the latent representation is akin to softmax being a relaxation of the $\arg\max$ operator. While we do not adjust the temperature $\tau \in [0,\infty)$ of the softmax used in SimNorm in our experiments, it is useful to note that it provides a mechanism for interpolating between two extremes. For example, $\tau \rightarrow \infty$ would force all probability mass onto single categories, resulting in the discrete codes (one-hot encodings) of VQ-VAE. The alternative of $\tau = 0$ would result in trivial codes (constant vectors; uniform probability mass) and prohibit propagation of information. SimNorm thus biases representations towards sparsity without enforcing discrete codes or other hard constraints.
> >
> > ----
> >
> > We have also addressed your other minor comments, such as the missing $\gamma$ below Eq. 3. Thank you for pointing this out! Please do not hesitate to let us know if you have any additional comments.

---

> ### Comment · Reviewer_BqZp · 2023-11-22
>
> I thank the authors for their detailed responses to my questions and commentary. I have no further concerns, and hence I am maintaining my decision of acceptance of the paper.

---

### Author Response · Authors · 2023-11-21
**General response and appreciation**

We thank all reviewers for their thoughtful comments; we really appreciate your feedback. We have revised our manuscript based on your feedback – the list of changes are available below. We have also responded to your individual comments.

**Summary of revisions:** We summarize changes to our manuscript below; these changes have also been highlighted (green) in the new version.

- Added new ablations on the choice of activation function as suggested by reviewer [BqZp](https://openreview.net/forum?id=Oxh5CstDJU&noteId=xCoQaNp0RE) (Appendix F). Mish activations produce smoother gradients overall, but TD-MPC2 works with any of the tested activations.

- Added new experiments on the benefit of offline RL techniques in our multi-task experiments as suggested by reviewer [8R3o](https://openreview.net/forum?id=Oxh5CstDJU&noteId=iXsUsifOSA) (Appendix J). Notably, existing offline RL techniques require task-specific tuning and are thus not readily applicable to our massively multi-task problem setting. Instead, we propose a novel test-time regularizer based on model uncertainty that requires no tuning (as with any of our other contributions), and find that it generally leads to better results.

- Added new experiments on scaling of multi-task models on a smaller task set (15 tasks) as suggested by reviewer [8R3o](https://openreview.net/forum?id=Oxh5CstDJU&noteId=iXsUsifOSA) (Appendix K).

- Added a new section on the main challenges in extending TD-MPC to discrete action spaces as suggested by reviewer [BqZp](https://openreview.net/forum?id=Oxh5CstDJU&noteId=xCoQaNp0RE) (Appendix I). We suggest replacing MPC with MCTS when actions are discrete.

- Added a new subsection on the intuition behind SimNorm as suggested by reviewer [BqZp](https://openreview.net/forum?id=Oxh5CstDJU&noteId=xCoQaNp0RE) and [jMX1](https://openreview.net/forum?id=Oxh5CstDJU&noteId=VBZdoCwvAT) (Appendix H). Intuitively, it can be thought of as a relaxation (soft variant) of the discrete codes from VQ-VAE.

- Minor rewording and updated references in Section 3 (TD-MPC2) as suggested by reviewer [jMX1](https://openreview.net/forum?id=Oxh5CstDJU&noteId=VBZdoCwvAT).

Again, we thank the reviewers for their constructive feedback. We believe that all comments have been addressed in this revision, but are happy to address any further comments from reviewers.

Best,

Authors of TD-MPC2 (submission 559)

---

### Author Response · Authors · 2023-11-22
**Thank you for a productive discussion period!**

As the discussion period is coming to an end, we want to thank all reviewers for the productive discussions that we have had over the past week. We are delighted to see that all reviewers share our enthusiasm for the paper, and genuinely believe that your feedback has helped strengthen it further.

Best,

Authors of TD-MPC2 (submission 559)

---

### Meta-Review · Area_Chair_eDSU · 2023-12-08

**Metareview:**

(a) Summary
The paper introduces TD-MPC2, an advancement of the TD-MPC model-based reinforcement learning algorithm. It incorporates novel architectural designs, employs maximum-entropy RL as the policy prior, and features a multi-task world model for learning various task dynamics. These improvements significantly enhance performance over existing state-of-the-art model-based and model-free algorithms in multi-task environments, including tasks from DM-Control and Meta-World. The authors provide detailed experimental results, open-source code, and a comprehensive discussion on the method's potential and limitations, addressing its adaptability and applicability in model-based reinforcement learning.

(b) Strength
This paper makes a significant contribution to model-based reinforcement learning by enhancing the TD-MPC algorithm's versatility and efficiency across multiple tasks. Its experimental results demonstrate substantial performance improvements over current leading methods. The paper is well-structured, offering a thorough presentation and open-source resources, which aid in further research and validation. Its strengths in detailed experimentation, clarity of presentation, and substantial advancements in model-based RL are underscored by the reviewers' appreciation of the method's robustness in complex multi-task scenarios and the comprehensive response to reviewers' questions.

(c) Weaknesses
It inherits certain drawbacks from MPC, such as computational overhead and difficulty with multi-modal transitions. I Concerns about the novelty of its modifications, which mainly derive from existing methods, and its mixed performance improvements in single-task scenarios compared to the original TD-MPC, may raise questions about its overall impact and innovation. Making it work for both continuous and discrete action spaces could be an interesting improvement.

**Justification For Why Not Higher Score:**

Although the paper is strong and presents valuable contributions to the field of model-based reinforcement learning, technically it could also be criticized that many improvements are by adapting existing techniques.

**Justification For Why Not Lower Score:**

TD-MPC is a prominent MBRL method for continuous control. Therefore, its significant advancements, particularly the notable performance improvements in multi-task reinforcement learning scenarios, have the potential to have a broad influence in the field. As a result, they are worth highlighting at the conference.

---

### Decision · Program_Chairs · 2024-01-16

Accept (spotlight)